# Simplicity bias and optimization threshold in two-layer ReLU networks

## Abstract

Understanding generalization of overparametrized neural networks remains a fundamental challenge in machine learning. Most of the literature mostly studies generalization from an interpolation point of view, taking convergence of parameters towards a global minimum of the training loss for granted. While overparametrized architectures indeed interpolated the data for typical classification tasks, this interpolation paradigm does not seem valid anymore for more complex tasks such as in-context learning or diffusion. Instead for such tasks, it has been empirically observed that the trained models goes from global minima to spurious local minima of the training loss as the number of training samples becomes larger than some level we call *optimization threshold*. While the former yields a poor generalization to the true population loss, the latter was observed to actually correspond to the minimiser of this true loss. This paper explores theoretically this phenomenon in the context of two-layer ReLU networks. We demonstrate that, despite overparametrization, networks often converge toward simpler solutions rather than interpolating the training data, which can lead to a drastic improvement on the test loss with respect to interpolating solutions. Our analysis relies on the so called early alignment phase, during which neurons align towards specific directions. This directional alignment, which occurs in the early stage of training, leads to a simplicity bias, wherein the network approximates the ground truth model without converging to the global minimum of the training loss. Our results suggest that this bias, resulting in an optimization threshold from which interpolation is not reached anymore, is beneficial and enhances the generalization of trained models.

## 1 Introduction

Understanding the generalization capabilities of neural networks remains a fundamental open question in machine learning (Zhang et al., 2021; Neyshabur et al., 2017). Traditionally, research has focused on explaining why neural networks models can achieve zero training loss while still generalizing well to unseen data in supervised learning tasks (Chizat and Bach, 2018; Mei et al., 2018; Rotskoff and Vanden-Eijnden, 2022; Chizat and Bach, 2020; Boursier et al., 2022; Boursier and Flammarion, 2023). This phenomenon is often attributed to overparametrization enabling models to find solutions that interpolate the training data yet avoid overfitting (Belkin et al., 2019; Bartlett et al., 2021).

However, the advent of generative AI paradigms—such as large language models (Vaswani et al., 2017) and diffusion models (Dhariwal and Nichol, 2021)—has introduced a paradigm shift in our understanding of generalization. In these settings, models can generate new data and perform novel tasks without necessarily interpolating the training data, raising fresh questions about how and why they generalize. To illustrate this shift, we consider two seemingly unrelated examples: in-context learning with transformers and generative modeling using diffusion methods.

Firstly, in-context learning (ICL) refers to the ability of large pretrained transformer models to learn new tasks from just a few examples, without any parameter updates (Brown et al., 2020; Min et al., 2022). A central question is whether ICL enables models to learn tasks significantly different from those encountered during pretraining. While prior work suggests that ICL leverages mechanisms akin to Bayesian inference (Xie et al., 2022; Garg et al., 2022; Bai et al., 2023), the limited diversity of tasks in pretraining datasets may constrain the model's ability to generalize. Raventós et al. (2024)

investigated this effect by focusing on regression problems to quantify how increasing the variety of tasks during pretraining affects ICL's capacity to generalize to new, unseen tasks, in context.

Secondly, diffusion models have made remarkable strides in generating high-quality images from high-dimensional datasets (Sohl-Dickstein et al., 2015; Ho et al., 2020; Song et al., 2021). These models learn to generate new samples by training denoisers to estimate the score function—the gradient of the log probability density—of the noisy data distribution (Song and Ermon, 2019). A significant challenge in this context is approximating a continuous density from a relatively small training set without succumbing to the curse of dimensionality. Although deep neural networks may tend to memorize training data when the dataset is small relative to the network's capacity (Somepalli et al., 2023; Carlini et al., 2023), Yoon et al. (2023); Kadkhodaie et al. (2023) observed they generalize well when trained on sufficiently large datasets, rendering the model's behavior nearly independent of the specific training set.

The common thread connecting these examples is a fundamental change in how gradient descent behaves in overparameterized models when the number of data points exceeds a certain threshold. Rather than converging to the global minimum of the training loss, gradient descent converges to a simpler solution closely related to the true loss minimizer. In learning scenarios involving noisy data, the most effective solutions are often those that do not interpolate the data. Despite their capacity to overfit, these models exhibit a simplicity bias, generalizing well to the underlying ground truth instead of merely fitting the noise in the training data. While simplicity bias generally refers to the tendency of models to learn features of increasing complexity, until reaching data interpolation (Arpit et al., 2017; Rahaman et al., 2019; Kalimeris et al., 2019; Huh et al., 2021); this phenomenon seems to stop before full interpolation in the case of in-context learning and diffusion models (even when training for a very long time). This observation underscores a significant shift in our understanding and approach to generalization in machine learning.

In this paper, we investigate this phenomenon in the setting of shallow ReLU networks applied to a regression problem. While multilayer perceptrons are foundational elements shared by the aforementioned models, focusing on shallow networks remains a significant simplification with respect to the architectures and algorithms used for training transformers and diffusion models. Despite this simplification, we aim to gain theoretical insights that could shed light on similar behaviors observed in more complex models.

Some recent works argued that overparametrized networks do not necessarily converge to global minima. In particular, Qiao et al. (2024) showed this effect for unidimensional data by illustrating the instability of global minima. Boursier and Flammarion (2024) advanced a different reason for this effect, given by the *early alignment* phenomenon: when initialized with sufficiently small weights, neurons primarily adjust their directions rather than their magnitudes in the early phase of training, aligning along specific directions determined by the stationary points of a certain function. This function can be explicitly characterized in simple cases.

**Contributions.**   Our first contribution is to show that this function driving the early alignment phase concentrates around its expectation, which corresponds to the true loss function. For simple teacher architectures, this expected function possesses only a few critical points. As a result, after the early alignment phase, the neurons become concentrated in a few key directions associated with the ground truth model. This behavior reveals a simplicity bias at the initial stages of training. Moreover, this directional concentration is believed to contribute to the non-convergence to the global minimizer of the training loss. However, this characterization only pertains to the initial stage of training. Therefore we extend our analysis to provide, under a restricted data model, a comprehensive characterization of the training dynamics, demonstrating that the simplicity bias persists until the end of training when the number of training samples exceeds some optimization threshold.

## 2 PRELIMINARIES

This section introduces the setting and the early alignment phenomenon, following the notations and definitions of Boursier and Flammarion (2024).

### 2.1 NOTATIONS

In the following, we denote by $\mathbb{S}_{d-1}$ the unit sphere of $\mathbb{R}^d$ and $B(\mathbf{0}, 1)$ the unit ball. We note $f(t) = \mathcal{O}_p\left(g(t)\right)$, if there exists a constant $C_p$, that only depends on $p$ such that for any $t$ in its

definition space, $|f(t)| \leq C_p g(t)$. We drop the $p$ index, if the constant $C_p$ is universal and does not depend on any parameter. Similarly in the Appendix, we note $f(t) = \Omega_p(g(t))$, if there exists a constant $C_p$, that only depends on $p$ such that for any $t$, $f(t) \geq C_p g(t)$. Moreover, we note $f(t) = \Theta_p(g(t))$ if both $f(t) = \mathcal{O}_p(g(t))$ and $f(t) = \Omega_p(g(t))$. For any bounded set $A$, $\mathcal{U}(A)$ denotes the uniform probability distribution on the set $A$.

## 2.2 SETTING

We consider $n$ data points $(x_k, y_k)_{k \in [n]} \in \mathbb{R}^{d+1}$ drawn i.i.d. from a distribution $\mu \in \mathcal{P}(\mathbb{R}^{d+1})$. We also denote by $\mathbf{X} = [x_1^\top, \ldots, x_n^\top] \in \mathbb{R}^{d \times n}$ and $\mathbf{y} = (y_1, \ldots, y_n) \in \mathbb{R}^n$ respectively the matrix whose columns are given by the input vectors $x_k$ and the vector with coordinates given by the labels $y_k$.

A two layer ReLU network is parameterised by $\theta = (w_j, a_j)_{j \in [m]} \in \mathbb{R}^{m \times (d+1)}$, corresponding to the prediction function

$$h_\theta : x \mapsto \sum_{j=1}^m a_j \sigma(w_j^\top x),$$

where $\sigma$ is the ReLU activation given by $\sigma(z) = \max(0, z)$. While training, we aim at minimizing the empirical square loss over the training data, defined as

$$L(\theta; \mathbf{X}, \mathbf{y}) = \frac{1}{2n} \sum_{k=1}^n (h_\theta(x_k) - y_k)^2.$$

As the limiting dynamics of (stochastic) gradient descent with vanishing learning rates, we study a subgradient flow of the training loss, which satisfies for almost any $t \in \mathbb{R}_+$,

$$\frac{\mathrm{d}\theta(t)}{\mathrm{d}t} \in -\partial_\theta L(\theta(t); \mathbf{X}, \mathbf{y}). \tag{1}$$

## 2.3 EARLY ALIGNMENT DYNAMICS

**Initialisation.** In accordance to the feature learning regime (Chizat et al., 2019), we consider a small initialisation scale, i.e., the $m$ neurons of the neural network are initialised as

$$(a_j(0), w_j(0)) = \frac{\lambda}{\sqrt{m}}(\tilde{a}_j, \tilde{w}_j), \tag{2}$$

where $\lambda > 0$ is the scale of initialisation and $(\tilde{a}_j, \tilde{w}_j)$ are vectors drawn i.i.d. from some distribution, such that they also satisfy the following domination property:

$$|\tilde{a}_j| \geq \|\tilde{w}_j\| \text{ for any } j \in [m] \qquad \text{and} \qquad \frac{1}{m} \sum_{j=1}^m \tilde{a}_j^2 \leq 1. \tag{3}$$

The domination property is common in the literature and allows for a simpler analysis, as it ensures that the signs of the output neurons $a_j(t)$ remain unchanged during training (Boursier et al., 2022).

**Neuron dynamics.** In the case of two layer neural networks with square loss and ReLU activation, Equation (1) can be written for each neuron $i \in [m]$ as

$$\frac{\mathrm{d}w_i(t)}{\mathrm{d}t} \in a_i(t)\mathfrak{D}_n(w_i(t), \theta(t))$$
$$\frac{\mathrm{d}a_i(t)}{\mathrm{d}t} = w_i(t)^\top D_n(w_i(t), \theta(t))\rangle, \tag{4}$$

where the vector $D_n(w_i(t), \theta(t))$ and set $\mathfrak{D}_n(w_i(t), \theta(t))$ are defined as follows, with $\partial\sigma$ the subdifferential of the activation $\sigma$,

$$D_n(w, \theta) = \frac{1}{n} \sum_{k=1}^n \mathbb{1}_{x_k^\top w > 0}(y_k - h_\theta(x_k))x_k,$$

$$\mathfrak{D}_n(w, \theta) = \left\{ \frac{1}{n} \sum_{k=1}^n \eta_k(y_k - h_{\theta(t)}(x_k))x_k \mid \eta_k \in \partial\sigma(x_k^\top w) \right\}.$$

These derivations directly follow from the subdifferential of the training loss. In particular, $D_n(w, \theta)$ corresponds to a specific vector (subgradient) in the subdifferential $\mathfrak{D}_n(w, \theta)$. Also observe that the set $\mathfrak{D}_n(w, \theta)$ depends on $w$ only through its activations $A_n(w)$, defined as

$$A_n: \begin{array}{l} \mathbb{S}_{d-1} \to \{-1, 0, 1\}^n \\ w \mapsto \text{sign}(w^\top x_k)_{k \in [n]} \end{array}.$$

Furthermore, $\mathfrak{D}_n(w, \theta)$ depends on $\theta$ only through the prediction function $h_\theta$, evaluated on the training inputs. This observation is crucial to the early alignment phenomenon. Notably, two neurons with the same activations follow similar dynamics.

**Early alignment.** In the small initialization regime described by Equation (2), numerous works highlight an early alignment phase in the initial stage of training (Maennel et al., 2018; Boursier and Flammarion, 2024; Kumar and Haupt, 2024; Tsoy and Konstantinov, 2024). During this phase, the neurons exhibit minimal changes in norm, while undergoing significant changes in direction. This phenomenon is due to a discrepancy in the derivatives of the neurons' norms (which scale with $\lambda$) and of their directions (which scale in $\Theta(1)$). Specifically, for a sufficiently small initialisation scale $\lambda$, the neurons align towards the critical directions of the following function $G_n$ defined on the $d$-dimensional sphere

$$G_n: \begin{array}{l} \mathbb{S}_{d-1} \to \mathbb{R} \\ w \mapsto \langle w, D_n(w, \mathbf{0}) \rangle \end{array}. \tag{5}$$

$G_n$ is a continuous, piecewise linear function which can be interpreted as the correlation between the gradient information around the origin and the neuron $w$ (given by $D_n(w, \mathbf{0})$). The network neurons thus align with the critical directions of $G_n$ during the early training dynamics. These critical directions are called *extremal vectors*, defined as follows.

**Definition 1.** *A vector $D \in \mathbb{R}^d$ is said **extremal** with respect to $G_n$ if there exists $w \in \mathbb{S}_{d-1}$ such that both simultaneously hold*

$$1. \quad D \in \mathfrak{D}_n(w, \mathbf{0}); \qquad 2. \quad D = \mathbf{0} \text{ or } A_n(D) \in \{A_n(w), -A_n(w)\}.$$

This definition directly follows from the KKT conditions of the maximization (or minimization) problem, constrained on the sphere, of the function $G_n$.

**Implications of early alignment.** By the end of the early alignment, most if not all neurons are nearly aligned with some extremal vector $D$. Maennel et al. (2018); Boursier and Flammarion (2024) argue that only a few extremal vectors exist in typical learning models. We further explore this claim in Section 3. As a consequence, only a few directions are represented by the network's weights at the end of the early dynamics, even though the neurons cover all possible directions at initialization. Boursier and Flammarion (2024) even show that this *quantization of directions* can prevent the network from interpolating the training set at convergence despite the overparametrization of the network.

Although this *failure of interpolation* has been considered a drawback by Boursier and Flammarion (2024), we show in Section 4 that it can also lead to a beneficial phenomenon of simplicity bias. Specifically, Section 4 illustrates on a simple linear example that for a large number of training samples, the model does not converge to interpolation. Instead, it converges towards the ordinary least square (OLS) estimator of the data. As a consequence, the model fits the true signal of the data, while effectively ignoring label noise. Before studying this example, we must first understand how extremal vectors behave as the number of training samples increases.

## 3 GEOMETRY OF ALIGNMENT IN THE LARGE SAMPLE REGIME

In this section, we aim to describe the geometry of the function $G_n$, with a specific focus on the extremal vectors, as the number of training samples $n$ becomes large. These vectors are key in driving the early alignment phase of the training, making them essential to understanding the initial dynamics of the parameters. Our approach involves first analyzing the concentration of gradient information $D_n$ of the train loss and then refining the analysis to focus on the extremal vectors.

Despite non-smoothness of the loss (due to ReLU activations), we can leverage the piecewise constant structure of the vector function $D_n(w)$, along with typical Rademacher complexity arguments, to derive uniform concentration bounds on the random function $w \mapsto D_n(w)$.

**Theorem 1.** *If the marginal law of $x$ is continuous with respect to the Lebesgue measure, then for any $n \in \mathbb{N}$,*

$$\mathbb{E}_{\mathbf{X},\mathbf{y}}\left[\sup_{w \in \mathbb{S}_{d-1}} \sup_{D_n \in \mathfrak{D}_n(w,\mathbf{0})} \|D_n - D(w)\|_2\right] = \mathcal{O}\left(\sqrt{\frac{d \log n}{n}} \mathbb{E}_\mu[\|\|yx\|_2^2]\right),$$

*where for any $w \in \mathbb{S}_{d-1}$, $D(w) = \mathbb{E}_\mu[\mathbb{1}_{w^\top x > 0} yx]$.*

Theorem 1 indicates that as $n$ grows large, the sets $\mathfrak{D}_n(w,\mathbf{0})$ converge to the corresponding vectors for the true loss, given by $D(w)$, at a rate $\sqrt{\frac{d \log n}{n}}$. Moreover this rate holds uniformly across all possible directions of $\mathbb{R}^d$ in expectation. A probability tail bound version of Theorem 1, which bounds this deviation with high probability, can also be derived (see Theorem 3 in Appendix C). A complete proof of Theorem 1 is provided in Appendix B.

When $n \to \infty$, the alignment dynamics are thus driven by vectors $D_n$ which are close to their expected value $D(w)$. Furthermore, when $n \to \infty$, the activations of a weight $A_n(w)$ exactly determine the direction of this weight, as every possible direction is then covered by the training inputs $x_k$. Specifically, for an infinite dataset with dense support indexed by $\mathbb{N}$, and defining the infinite activation function $A$ as

$$A: \begin{array}{l} \mathbb{S}_{d-1} \to \{-1, 0, 1\}^{\mathbb{N}} \\ w \mapsto \text{sign}(w^\top x_k)_{k \in \mathbb{N}} \end{array} ;$$

then $A$ is injective. In this infinite data limit, the functions $G_n$ converge to the differentiable function $G: w \mapsto w^\top D(w)$ and a vector $D \in \mathbb{R}^d$ is **extremal** with respect to $G$ if there exists $w \in \mathbb{S}_{d-1}$ such that both

$$1. \quad D = D(w) \qquad 2. \quad D = \mathbf{0} \text{ or } \frac{D}{\|D\|_2} \in \{w, -w\}. \tag{6}$$

When $n$ becomes large, the extremal vectors of the data then concentrate toward the vectors satisfying Equation (6). This is precisely quantified by Proposition 1 below.

**Proposition 1.** *Assume the marginal law of $x$ is continuous with respect to the Lebesgue measure and that $\mathbb{E}[\|xy\|^4] < \infty$.*

*Then for any $\varepsilon > 0$, there is $n^\star(\varepsilon) = \mathcal{O}_{\varepsilon,\mu}(d \log d)$ such that for any $n \geq n^\star(\varepsilon)$, with probability at least $1 - \mathcal{O}_\mu\left(\frac{1}{n}\right)$: for any extremal vector $D_n$ of the finite data $(\mathbf{X}, \mathbf{y}) \in \mathbb{R}^{n \times (d+1)}$, there exists a vector $D^\star \in \mathbb{R}^d$ satisfying Equation (6), such that*

$$\|D_n - D^\star\|_2 \leq \varepsilon.$$

Proposition 1 states that for large $n$, the extremal vectors concentrate towards the vectors satisfying Equation (6). The proof of Proposition 1 relies on the tail bound version of Theorem 1 and continuity arguments. A complete proof is given in Appendix D.

**Early alignment towards a few directions.**    Besides laying the ground for Theorem 2, Proposition 1 aims at describing the geometry of the early alignment when the number of training samples grows large. In particular, Proposition 1 shows that all extremal vectors concentrate towards the directions satisfying Equation (6). Although such a description remains abstract, we believe it is satisfied by only a few directions for many data distributions. As an example, for symmetric data distributions, it is respectively satisfied by a single or two directions, when considering a one neuron or linear teacher. More generally, we conjecture it should be satisfied by a small number of directions as soon as the labels are given by a small teacher network. Proving such a result is yet left for future work.

The early alignment phenomenon has been described in many works, to show that after the early training dynamics, only a few directions (given by the extremal vectors) are represented by the neurons (Bui Thi Mai and Lampert, 2021; Lyu et al., 2021; Boursier et al., 2022; Chistikov et al., 2023; Min et al., 2024; Boursier and Flammarion, 2024; Tsoy and Konstantinov, 2024). However, these works all rely on specific data examples, where extremal vectors can be easily expressed for a finite number of samples. Proposition 1 aims at providing a more general result, showing that for large $n$, it is sufficient to consider the directions satisfying Equation (6), which is easier to characterize from a statistical perspective. We thus believe that Proposition 1 advances our understanding of how sparse is the network representation (in directions) at the end of early alignment.

Proposition 1 implies that for large values of $n$ ($\gtrsim d$), the early alignment phase results in the formation of a small number of neuron clusters, effectively making the neural network equivalent to a small-width network. Empirically, these clusters appear to be mostlypreserved throughout training. The neural network then remains equivalent to a small-width network along its entire training trajectory.

In contrast, when the number of data is limited ($n \lesssim d$), this guarantee no longer holds and a large number of extremal vectors may exist. For example in the case of orthogonal data (which only holds for $n \leq d$), there are $\Theta(2^n)$ extremal vectors (Boursier et al., 2022). In such cases, there would still be a large number of neuron clusters at the end of the early alignment phase, maintaining a large *effective width* of the network. Studying how this effective width is maintained until the end of training in the orthogonal case remains an open problem. We conjecture that for a mild overparametrization ($n \lesssim m \ll 2^n$),[1] we would still have a relatively large effective width (increasing with $n$) at the end of training.

## 4 OPTIMISATION THRESHOLD AND SIMPLICITY BIAS

The goal of this section is to illustrate the transition from interpolating the training data to a nearly optimal estimator (with respect to the true loss) that can arise when increasing the size of training data. Toward this end, this section proves on a simple data example, that for a large enough number of training samples, an overparametrized network will not converge to a global minimum of the training loss, but will instead be close to the minimizer of the true loss.

More precisely, we consider the specific case of a linear data model:

$$y_k = x_k^\top \beta^\star + \eta_k \quad \text{for any } k \in [n], \tag{7}$$

where $\eta_k$ is some noise, drawn i.i.d. from a centered distribution. We also introduce a specific set of assumptions regarding the data distribution.

**Assumption 1.** *The samples $x_k$ and the noise $\eta_k$ are drawn i.i.d. from distributions $\mu_X$, and $\mu_\eta$ satisfying, for some $c > 0$:*

1. *$\mu_X$ is symmetric, i.e., $x_k$ and $-x_k$ follow the same distribution;*

2. *$\mu$ is continuous with respect to the Lebesgue measure;*

3. *$\mathbb{P}_{x \sim \mu_X} \left( |x^\top \beta^\star| \leq c \frac{\|x\|_2}{\sqrt{d}} \right) = 0$;*

4. *$\|\mathbb{E}_{x \sim \mu_X}[xx^\top] - \mathbf{I}_d\|_{op} < \min \left( \frac{c}{2\sqrt{d}\|\beta^\star\|_2}, \frac{3}{5} \right)$;*

5. *The random vector $x_k$ is 1 sub-Gaussian and the noise satisfies $\mathbb{E}_\eta[\eta^4] < \infty$.*

Conditions 1, 2 and 5 in Assumption 1 are relatively mild. However, item 3 is quite restrictive: it is needed to ensure that the volume of the activation cone containing $\beta^\star$ does not vanish when $n \to \infty$. A similar assumption is considered by Chistikov et al. (2023); Tsoy and Konstantinov (2024), for similar reasons. Additionally, Condition 4 ensures that $\mathbb{E}_x[xx^\top]\beta^\star$ and $\beta^\star$ are in the same activation cone. This assumption allows the training dynamics to remain within a single cone after the early alignment phase, significantly simplifying our analysis.

As an example, note that if the samples $x_k$ are distributed i.i.d. as

$$x_k = \mathsf{s}_k \frac{\beta^\star}{\|\beta^\star\|} + \sqrt{d-1}\mathsf{v}_k \quad \text{with}$$

$$\mathsf{s}_k \sim \mathcal{U}\left([-1-\varepsilon, -1+\varepsilon] \cup [1-\varepsilon, 1+\varepsilon]\right) \text{ and } \mathsf{v}_k \sim \mathcal{U}(\mathbb{S}_{d-1} \cap \{\beta^\star\}^\perp),$$

for a small enough $\varepsilon > 0$ and $\mu_\eta$ a standard Gaussian distribution, then Assumption 1 is satisfied. In this section, we also consider the following specific initialisation scheme for any $i \in [m]$:

$$w_i(0) \sim \frac{\lambda}{2\sqrt{m}}\mathcal{U}(B(\mathbf{0}, 1)) \quad \text{and} \quad a_i(0) \sim \frac{\lambda}{\sqrt{m}}\mathcal{U}(\{-1, 1\}). \tag{8}$$

---

[1]Boursier et al. (2022) proved an effective width of 2 at the end of training when $m \gtrsim 2^n$.

In addition to the regime considered in Equations (2) and (3), this initialization introduces a stronger domination condition, as $|a_i(0)| \geq 2\|w_i(0)\|$. This condition reinforces the early alignment phase, ensuring that **all** neurons are nearly aligned with extremal vectors by the end of this phase. Assumption 1 and Equation (8) are primarily introduced to enable a tractable analysis and are discussed further in Section 4.2.

This set of assumptions allows to study the training dynamics separately on the following partition of the data:

$$\mathcal{S}_+ = \{k \in [n] \mid x_k^\top \beta^\star \geq 0\} \quad \text{and} \quad \mathcal{S}_- = \{k \in [n] \mid x_k^\top \beta^\star < 0\}.$$

Hereafter, we denote by $\mathbf{X}_+ \in \mathbb{R}^{d \times |\mathcal{S}_+|}$ (resp. $\mathbf{X}_-$), the matrix with rows given by the vectors $x_k$ for $k \in \mathcal{S}_+$ (resp. $k \in \mathcal{S}_-$). Similarly, we denote by $\mathbf{Y}_+ \in \mathbb{R}^{|\mathcal{S}_+|}$ (resp. $\mathbf{Y}_-$) the vector with coordinates given by the labels $y_k$ for $k \in \mathcal{S}_+$ (resp. $k \in \mathcal{S}_-$).

Studying separately the positive ($a_i > 0$) and negative ($a_i < 0$) neurons, we prove Theorem 2 below, which states that at convergence and for a large enough number of training samples, the sum of the positive (resp. negative) neurons correspond to the OLS estimator on the data subset $\mathcal{S}_+$ (resp. $\mathcal{S}_-$).

**Theorem 2.** *If Assumption 1 holds and the initialisation scheme follows Equation* (8)*, then there exists $\lambda^\star = \Theta(\frac{1}{d})$ and $n^\star = \Theta(d^3 \log d)$ such that for any $\lambda \leq \lambda^\star$, any $m \in \mathbb{N}$ and $n \geq n^\star$, with probability $1 - \mathcal{O}\left(\frac{d^2}{n} + \frac{1}{2^m}\right)$, the parameters $\theta(t)$ converge to some $\theta_\infty$ such that*

$$h_{\theta_\infty}(x) = (\beta_{n,+}^\top x)_+ - (-\beta_{n,-}^\top x)_+ \quad \text{for any } x \in \text{Supp}(\mu_X),$$

*where $\text{Supp}(\mu_X)$ is the support of the distribution $\mu_X$, $\beta_{n,+} = (\mathbf{X}_+ \mathbf{X}_+^\top)^{-1} \mathbf{X}_+ \mathbf{Y}_+$ and $\beta_{n,-} = (\mathbf{X}_- \mathbf{X}_-^\top)^{-1} \mathbf{X}_- \mathbf{Y}_-$ are the OLS estimator respectively on the data in $\mathcal{S}_+$ and $\mathcal{S}_-$.*

Precisely, the estimator learnt at convergence for a large enough $n$ behaves $\mu_X$-everywhere as the difference of two ReLU neurons, with nearly opposite directions (thanks to the distribution symmetry), resulting in a nearly linear estimator. These directions correspond to the OLS estimator of the data in $\mathcal{S}_+$ and in $\mathcal{S}_-$, respectively.

The complete proof of Theorem 2 is deferred to Appendix E. We provide a detailed sketch in Section 4.1 below and discuss further Theorem 2 in Section 4.2.

### 4.1 SKETCH OF PROOF OF THEOREM 2

The proof of Theorem 2 examines the complete training dynamics of positive neurons ($a_i(0) > 0$) and negative ones ($a_i(0) < 0$) separately. This decoupling is possible at the end of the early phase, due to Assumption 1, and is handled thanks to Lemma 4 in the Appendix.

First note that for the given model, there are only two vectors satisfying Equation (6), corresponding to $\frac{1}{2}\Sigma\beta^\star$ and $-\frac{1}{2}\Sigma\beta^\star$ respectively, for $\Sigma = \mathbb{E}_{x \sim \mu_X}[xx^\top]$. From then and thanks to the third point of Assumption 1, the results from Section 3 imply that, for a large value of $n$ and with high probability, there are only two extremal vectors, both of which are close to the expected ones mentioned above.

By analysing the early alignment phase similarly to Boursier and Flammarion (2024), we show that by the end of this early phase, (i) all neurons have small norms; (ii) positive (resp. negative) neurons are aligned with $\Sigma\beta^\star$ (resp. $-\Sigma\beta^\star$). More specifically, at time $\tau$, defined as the end of the early alignment phase, we show that

$$\forall i \in [m], \quad \frac{w_i(\tau)}{a_i(\tau)}^\top \Sigma\beta^\star = \|\Sigma\beta^\star\| - \mathcal{O}\left(\lambda^\varepsilon + \sqrt{\frac{d^2 \log n}{n}}\right).$$

From that point onward, all positive neurons are nearly aligned and behave as a single neuron until the end of training. Moreover, they remain in the same activation cone until the end of training. Namely for any $i \in [m]$ and $t \geq \tau$,

$$\text{sign}(a_i(t))\langle w_i(t), x_k \rangle > 0 \quad \text{for any } k \in \mathcal{S}_+,$$
$$\text{sign}(a_i(t))\langle w_i(t), x_k \rangle < 0 \quad \text{for any } k \in \mathcal{S}_-.$$

We then show that during a second phase, all positive neurons grow until they reach the OLS estimator on the data in $\mathcal{S}_+$. Mathematically, for some time $\tau_{2,+} > \tau$,

$$\sum_{i, a_i(0) > 0} a_i(\tau_{2,+}) w_i(\tau_{2,+}) \approx \beta_{n,+}.$$

Similarly, negative neurons end up close to $\beta_{n,-}$ after a different time $\tau_{2,-}$. Proving this second phase is quite technical and is actually decomposed into a slow growth and fast growth phases, following a similar approach to Lyu et al. (2021); Tsoy and Konstantinov (2024).

At the end of the second phase, the estimation function is already close to the one described in Theorem 2. From then, we control the neurons using a local Polyak-Łojasiewicz inequality (see Equation 45) to show that they remain close to their value at the end of the second phase, and actually converge to a local minimum corresponding to the estimation function $h_{\theta_\infty}$ described in Theorem 2.

### 4.2 DISCUSSION

This section discusses in details Theorem 2 and its limitations.

**Absence of interpolation.** For many years, the literature has argued in favor of the fact that, if overparametrized enough, neural networks do converge towards interpolation of the training set, i.e., to a global minimum of the loss (Jacot et al., 2018; Du et al., 2019; Chizat and Bach, 2018; Wojtowytsch, 2020).

Yet, some recent works argued in the opposite direction that convergence towards global minima might not be achieved for regression tasks, even with infinitely overparametrized networks (Qiao et al., 2024; Boursier and Flammarion, 2024). Indeed, Theorem 2 still holds as $m \to \infty$: although interpolation of the data is possible from a statistical aspect, interpolation does not occur for optimization reasons. In this direction, Qiao et al. (2024) claim that for large values of $n$ and univariate data, interpolation cannot happen because of the large (i.e., finite) stepsizes used for gradient descent. Following Boursier and Flammarion (2024), we here provide a complementary reason, which is due to the early alignment phenomenon and loss of omnidirectionality of the weights (i.e., the fact that the weights represent all directions in $\mathbb{R}^d$). Note that this loss of omnidirectionality is specific to the (leaky) ReLU activation and does not hold for smooth activations (see e.g. Chizat and Bach, 2018, Lemma C.10). We experimentally confirm in Appendix A.5 that both visions are complementary, as interpolation still does not happen for arbitrarily small learning rates.

**Simplicity bias.** Simplicity bias has been extensively studied in the literature (Arpit et al., 2017; Rahaman et al., 2019; Kalimeris et al., 2019; Huh et al., 2021). It is often described as the fact that networks learn features of increasing complexity while learning. In other words, simpler features are first learnt (e.g., a linear estimator), and more complex features might be learnt later. This has been observed in many empirical studies, leading to improved performance in generalization, except from a few nuanced cases (Shah et al., 2020). Yet in all these studies, the network interpolates the training set after being trained for a long enough time. In consequence, simplicity bias has been characterized by a first *feature learning phase*; and is then followed by an *interpolating phase*, where the remaining noise is fitted (Kalimeris et al., 2019).

We here go further by showing that this last interpolating phase does not even happen in some cases. Theorem 2 indeed claims that after the first feature learning phase, where the network learns a linear estimator, nothing happens in training. The interpolating phase never starts, no matter how long we wait for. While interpolation is often observed for classification problems in practice, it is generally much harder to reach for regression problems (Stewart et al., 2022; Yoon et al., 2023; Kadkhodaie et al., 2023; Raventós et al., 2024). Theorem 2 confirms this tendency by illustrating a regression example where interpolation does not happen at convergence.

Although implicit bias and simplicity bias often refer to the same behavior in the literature, we here distinguish the two terms: implicit bias is generally considered in the regime of interpolation (Soudry et al., 2018; Lyu and Li, 2019; Chizat and Bach, 2020; Ji and Telgarsky, 2019), while simplicity bias still exists in absence of interpolation.

**Improved test loss, due to overparametrization threshold.** Theorem 2 states that for a large enough number of training samples, the interpolating phase does not happen during training, and

the estimator then resembles the OLS estimator of the training set. In that regime, the excess risk scales as $\mathcal{O}\left(\frac{d}{n}\right)$ (Hsu et al., 2011) and thus quickly decreases to 0 as the number of training samples grows. In contrast when interpolation happens, we either observe a *tempered overfitting*, where the excess risk does not go down to 0 as the number of samples grows (Mallinar et al., 2022); or even a *catastrophic overfitting*, where the excess risk instead diverges to infinity as the size of the training set increases (Joshi et al., 2023).

The fact that the excess risk goes down to 0 as $n$ grows in our example of Section 4 could not be due to a benign overfitting (Belkin et al., 2018; Bartlett et al., 2020), as benign overfitting occurs when the dimension $d$ also grows to infinity. We here consider a fixed dimension instead, and this reduced risk is then solely due to the *optimization threshold*, i.e., the fact that for a large enough $n$, the interpolating phase does not happen anymore. While some works rely on early stopping before this interpolating phase to guarantee such an improved excess risk (Ji et al., 2021; Mallinar et al., 2022; Frei et al., 2023), it can be guaranteed without any early stopping after this optimization threshold. A similar threshold has been empirically observed in diffusion and in-context learning (Yoon et al., 2023; Kadkhodaie et al., 2023; Raventós et al., 2024), where the trained model goes from interpolation to generalization as the number of training samples increases.

**Limitations and generality.** While Theorem 2 considers a very specific setting, it describes a more general behavior. Although condition 3 of Assumption 1 and the initialization scheme of Equation (8) are quite artificial, they are merely required to allow a tractable analysis. The experiments of Section 5 are indeed run without these conditions and yield results similar to the predictions of Theorem 2 for large enough $n$.

More particularly, condition 3 of Assumption 1 is required to ensure that only two extremal vectors exist. Without this condition, there could be additional extremal vectors, but all concentrated around these two main extremal ones. On the other hand, Equation (8) is required to enforce the early alignment phase, so that all neurons are aligned towards extremal vectors at its end. With a more general initialization, some neurons could move arbitrarily slowly in the early alignment dynamics, ending unaligned at the end of early phase. Yet, such neurons would be very rare. Relaxing these two assumptions would make the final convergence point slightly more complex than the one in Theorem 2. Besides the two main ReLU components described in Theorem 2, a few small components could also be added to the final estimator, without significantly changing the reached excess risk, as observed in Section 5. This is observed in Figure 1, where the training loss is only slightly smaller than the training loss of OLS, with a comparable test loss.

From a higher level, Theorem 2 is restricted to a linear teacher and a simple network architecture. It remains hard to assess how well the considered setting reflects the behavior of more complex architectures encountered in practice. We believe that the different conclusions of our work remain valid in more complex setups. In particular, additional experiments in Appendix A run with a more complex teacher, GeLU activations or with Adam optimizer yield similar behaviors: the obtained estimator does not interpolate for a large number of training samples, but instead accurately approximates the minimizer of the test loss. Similar behaviors have also been observed on more complex tasks as generative modeling or in-context learning (Yoon et al., 2023; Kadkhodaie et al., 2023; Raventós et al., 2024). Despite overparametrization, the trained model goes from perfect interpolation to generalization, as it fails at interpolating for a large number of training samples. In these works as well, this absence of interpolation does not seem due to an early stopping, but rather to convergence to a local minimum (see e.g., Raventós et al., 2024, Figure 4).

Lastly, Theorem 2 requires a very large number of samples with respect to the dimension, i.e., $n \gtrsim d^3 \log d$. Our experiments confirm that the optimization threshold only appears for a very large number of training samples with respect to the dimension. However, similar behaviors seem to occur for smaller orders of magnitude for $n$ in more complex learning problems, such as the training of diffusion models (Yoon et al., 2023; Kadkhodaie et al., 2023). This dependency in $d$ might indeed be different for more complex architectures (e.g., with attention) and is worth investigating for future work.

## 5 EXPERIMENTS

This section illustrates our results on experiments on a toy model close to the setting of Section 4. More precisely, we train overparametrized two-layer neural networks ($m = 10\,000$) until convergence,

on data from the linear model of Equation (7). The network is trained via stochastic gradient descent and the dimension is here fixed to $d = 5$ to allow reasonable running times. The setup here is more general than Section 4, since i) the data input $x_k$ are drawn from a standard Gaussian distribution (which does not satisfy Assumption 1); ii) the neurons are initialized as centered Gaussian of variance $\frac{10^{-5}}{m}$ (which does not satisfy Equations (3) and (8)). We refer to Appendix A for details on the considered experiments and additional experiments.

Figure 1 illustrates the behavior of both train loss and test loss at convergence, when the size of the training set $n$ varies. As predicted by Theorem 2, when $n$ exceeds some optimization threshold, the estimator at convergence does not interpolate the training set. Instead, it resembles the optimal OLS estimator, which yields a test loss close to the noise level $\mathbb{E}[\eta^2]$. In contrast for smaller training sets, the final estimator interpolates the data at convergence, which yields a much larger test loss than OLS, corresponding to the tempered overfitting regime (Mallinar et al., 2022).

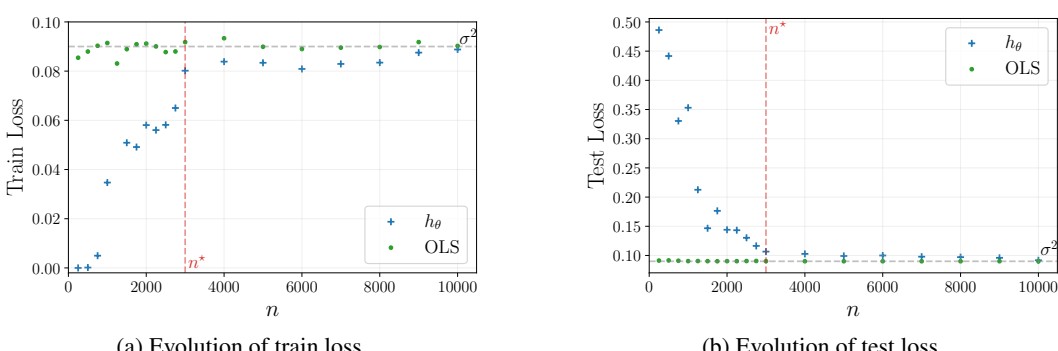

(a) Evolution of train loss.          (b) Evolution of test loss.

Figure 1: Evolution of both train and test losses at convergence with respect to the number of training samples. $\sigma^2$ corresponds to the noise variance $\mathbb{E}[\eta^2]$.

This optimization threshold is here located around $n^\star = 3000$, which suggests that the large dependency of this threshold in the dimension (which is here 5) in Theorem 2 seems necessary.[2]

We still observe a few differences here with the predictions of Theorem 2, which are due to the two differences in the setups mentioned above. Indeed, even after this optimization threshold, the test loss of the obtained network is slightly larger than the one of OLS, while Theorem 2 predicts they should coincide. This is because in the experimental setup, a few neurons remain disaligned with the extremal ones at the end of the early alignment phase. These neurons will then later in training grow in norm, trying to fit a few data points. However there are only a few of such neurons, whose impact thus becomes limited. As a consequence, they only manage to slightly improve the train loss, and thus only slightly degrade the test loss.

## 6    CONCLUSION

This work illustrates on a simple linear example the phenomenon of non-convergence of the parameters towards a global minimum of the training loss, despite overparametrisation. This non-convergence actually yields a simplicity bias on the final estimator, which can lead to an optimal fit of the true data distribution. A similar phenomenon has been observed on more complex and realistic settings (Yoon et al., 2023; Kadkhodaie et al., 2023; Raventós et al., 2024), for which a theoretical analysis remains intractable.

This result is proven via the description of the early alignment phase. Besides the specific data example considered in Section 4, we also provide concentration bounds on the extremal vectors driving this early alignment. We believe this result can be used in subsequent works to better understand this early phase of the training dynamics, and how it yields biases towards simple estimators.

---

[2]See Appendix A.6 for experiments with larger dimensions.

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

# Appendix

## Table of Contents

## A ADDITIONAL EXPERIMENTS

### A.1 EXPERIMENTAL DETAILS

In the experiments of Figure 1, we initialised two-layer ReLU networks (without bias term) with $m = 10\,000$ neurons, initialized i.i.d. for each component as a Gaussian of variance $\frac{10^{-5}}{\sqrt{m}}$. We then generated training samples as

$$y_k = {\beta^\star}^\top x_k + \eta_k,$$

where $\eta_k$ are drawn i.i.d. as centered Gaussian of variance $\sigma^2 = 0.09$, $x_k$ are drawn i.i.d. as centered Gaussian variables and $\beta^\star$ is fixed, without loss of generality, to $\beta^\star = (1, 0, \ldots, 0)$. The dimension is fixed to $d = 5$. We then train these networks on training datasets of different sizes (each dataset is resampled from scratch).

The neural networks are trained via stochastic gradient descent (SGD), with batch size 32 and learning rate 0.01. To ensure that we reached convergence of the parameters, we train the networks for $8 \times 10^6$ iterations of SGD, where the training seems stabilized.

All the experiments were run on a personal MacBook Pro, for a total compute time of approximately 100 hours.

### A.2 COSINE SIMILARITY WITH OLS ESTIMATOR

To illustrate Theorem 2 and the fact that neurons end up aligned with the OLS estimator beyond the optimization threshold $n^\star$, Figure 2 shows histograms of the cosine similarities[3] between the all the neurons $w_i$ of the network at the end of training and the true OLS estimator $\hat{\beta} = (\mathbf{X}\mathbf{X}^\top)^{-1}\mathbf{X}\mathbf{Y}$, for

---

[3]The cosine similarity between two vectors $u, v \in \mathbb{R}^d$ is defined as $\cos(u, v) = \frac{u^\top v}{\|u\|\,\|v\|}$.

different sample complexities. This experiment follows the exact same setup as the one described in Appendix A.1.

In particular, Figure 2a shows this histogram for $n = 500$, where interpolation of the training data happens (see Figure 1a); and Figure 2b shows this histogram for $n = 5\,000$, where interpolation of the training data does not happen anymore, but the network generalizes well to unseen data.

While a majority of the neurons is already nicely aligned with the true OLS estimator in the former case (having a cosine similarity close to either $1$ or $-1$), we see that nearly all neurons are aligned with this true estimator as $n$ grows larger, confirming the predictions of Theorem 2. As explained in Section 5, there are still a few vectors that are disaligned with the OLS estimator here, but they have almost no impact on the estimated function.

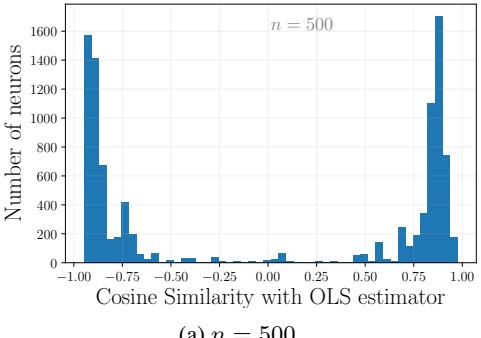
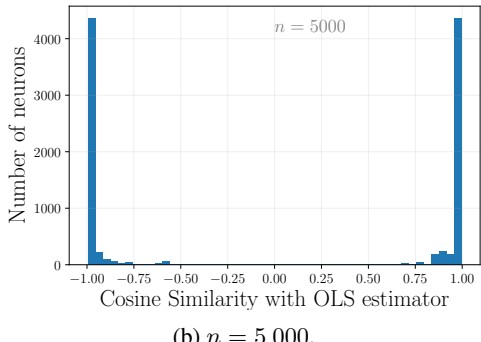

(a) $n = 500$.      (b) $n = 5\,000$.

Figure 2: Histogram of the cosine similarities of the neurons with the true OLS estimator $\hat{\beta}$, at the end of training.

### A.3 GeLU activation

Our theoretical results can be directly extended to any homogeneous activation function, i.e., leaky ReLU activation. Yet, the theory draws different conclusions for differentiable activations functions and claims that for infinitely wide neural networks, the parameters should interpolate the data at convergence (Chizat and Bach, 2018). This result yet only holds for infinitely wide networks, and it remains unknown how wide a network should be to actually reach such an interpolation in practice. Figure 3 below presents experiments similar to Section 5, replacing the ReLU activation by the differentiable GeLU activation (Hendrycks and Gimpel, 2016). This activation is standard in modern large language models. Notably, it is used in the GPT2 architecture, which was used in the experiments of Raventós et al. (2024).

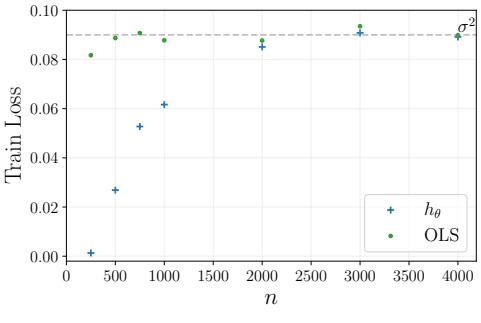
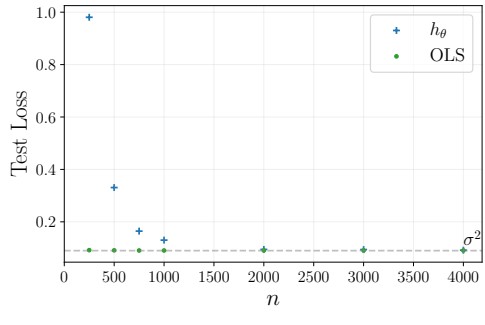

(a) Evolution of train loss.      (b) Evolution of test loss.

Figure 3: Evolution of both train and test losses at convergence with respect to the number of training samples, with GeLU activation.

While infinitely wide GeLU networks should overfit, even very wide networks ($m = 10\,000$) are far from this behavior in practice. In particular, we observe a phenomenon similar to Section 5 in Figure 3. Surprisingly, it even seems that interpolation is harder to reach with GeLU activation, as the network is already unable to interpolate for $n = 500$ training samples. We believe this is due to the

fact that GeLU is close to a linear function around the origin (corresponding to our small initialization regime), making it harder to overfit noisy labels.

## A.4 MOMENTUM BASED OPTIMIZERS

Our theoretical results hold for Gradient Flow, which is a first order approximation of typical gradient methods such as Gradient Descent (GD) or Stochastic Gradient Descent (SGD) (Li et al., 2019). Yet, recent large models implementations typically use different, momentum based algorithms, such as Adam (Kingma, 2014) or AdamW (Loshchilov, 2017). To illustrate the generality of the optimization threshold we proved in a specific theoretical setting, we consider in Figure 4 below the same experiments as in Section 5, with the exception that i) we used GeLU activation functions (as in Appendix A.3) and ii) we minimized the training loss through the Adam optimizer, with pytorch default hyperparameters.

We focus on Adam rather than AdamW here to follow the experimental setup of Raventós et al. (2024) and because our focus is on implicit regularization, thus avoiding explicit regularization techniques.

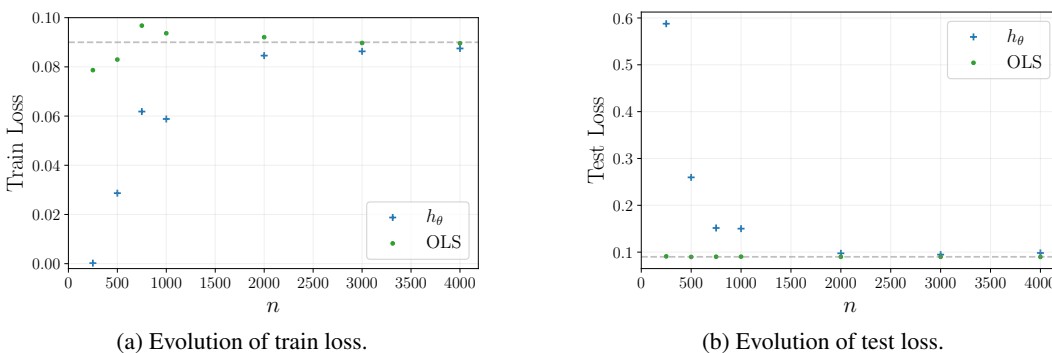

(a) Evolution of train loss.                    (b) Evolution of test loss.

Figure 4: Evolution of both train and test losses at convergence with respect to the number of training samples, with GeLU activation and Adam optimizer.

The observed results are very similar to the ones of Figure 3, leading to similar conclusions than Appendix A.3 and the fact that considering Adam rather than SGD does not significantly change the final results.

## A.5 STABILITY OF MINIMA

Qiao et al. (2024) argue that the non-convergence of the estimator towards interpolation is due to the instability of global minima. More precisely they claim that for large stepsizes, gradient descent (GD) cannot stabilize around global minima of the loss for large values of $n$. We present an additional experiment in this section, illustrating that this non-convergence is not due to an instability of the convergence point of (S)GD, but to it being a stationary point of the loss as predicted by our theory.

For that, we consider a neural network initialized from the final point (warm restart) of training for $8\,000$ samples in the experiment of Figure 1.[4] We then continue training this network on the same training dataset, with a decaying learning rate schedule. Precisely, we start with a learning rate of $0.01$ as in the main experiment, and multiply the learning rate by $0.85$ every $50\,000$ iterations of SGD, so that after $4 \times 10^6$ iterations, the final learning rate is of order $10^{-8}$.

We observe on Figure 5 that the training loss does not change much from the point reached at the end of training with the large learning rate $0.01$. Indeed, the training loss was around $0.082$ at the end of this initial training, which is slightly less than the noise level ($0.09$). While there seems to be some stabilization happening at the beginning of this decaying schedule, the training loss seems to converge to slightly more than $0.0815$, confirming that the absence of interpolation is not due to an instability reason, but rather to a convergence towards a spurious stationary point of the loss.

---

[4]Another relevant experiment is to train from scratch (no warm restart) with a smaller learning rate. When running the experiment of Appendix A.1 with a smaller learning rate $0.001$, we observe again that the parameters at convergence correspond to the OLS estimator.

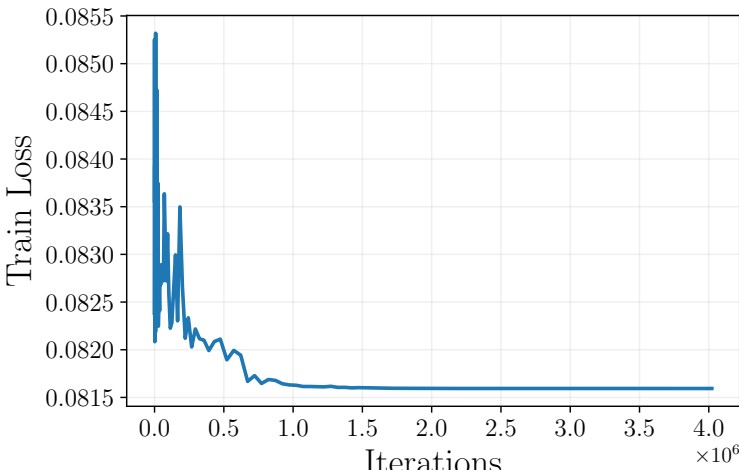

Figure 5: Evolution of training loss from warm restart with a decaying learning rate schedule ($n = 8000, d = 5$).

## A.6 INFLUENCE OF DIMENSIONALITY

Theorem 2 predicted an optimization threshold scaling in $\mathcal{O}(d^3 \log d)$. However, the experiments of Section 5 consider a fixed dimension ($d = 5$), making it unclear how tight is this theoretical optimization threshold and whether a similar dependency in the dimension is observed in practice. To investigate further this dependency in the dimension, we present in this section experiments in the same setup described in Appendix A.1, with the sole exception that the dimension is larger, fixed to $d = 10$.

Figure 6 illustrates the evolution of both the train and test losses as the number of training samples increases in this larger dimension setting. In that case, the optimization threshold seems much larger: interpolation stops happening around $n = 10\,000$ samples, and an estimation close to the OLS estimator really starts happening at much larger values of $n$, around $n = 80\,000$.

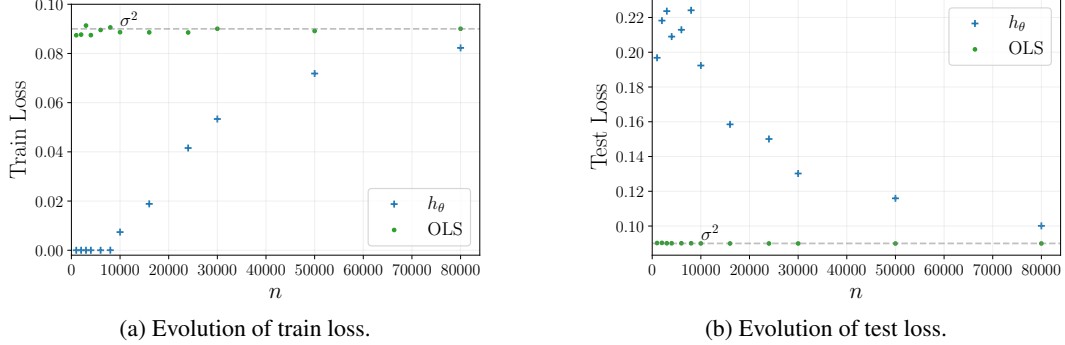

(a) Evolution of train loss.          (b) Evolution of test loss.

Figure 6: Evolution of both train and test losses at convergence with respect to the number of training samples, with dimension $d = 10$.

Comparing with the $d = 5$ case, it thus seems that the point at which interpolation stops indeed seems to roughly scale in $d^3$. However, this scaling seems even larger for the point where the estimator corresponds to the OLS one. We believe that this discrepancy is due to the differences between our theoretical and experimental setups, and in particular to the fact that multiple intermediate neurons can grow in our experimental setup (see *Limitations and generality* paragraph in Section 4.2).

## A.7 5 RELU TEACHER NETWORK

This section presents an additional experiment with a more complex data model. More precisely, we consider the exact same setup than Section 5 (described in Appendix A.1), with the difference that

the labels $y_k$ are given by

$$y_k = f^\star(x_k) + \eta_k,$$

where $f^\star$ is a 5 ReLU network:

$$f^\star(x_k) = \frac{1}{5}\sum_{i=1}^{5}(x_k^\top \beta_i^\star)_+.$$

The parameters $\beta_i^\star$ are drawn i.i.d. at random following a standard Gaussian distribution. We use the exact same $\beta_i^\star$ across all the runs for different values of $n$. Also, $x_k$ and $\eta_k$ are generated in the same way as described in in Appendix A.1.

Figure 7 also presents the evolution of the train and test losses as the number of training samples varies. We observe a behavior similar to Figure 1, where interpolation is reached for small values of $n$, and is not reached anymore after some threshold $n^\star$. While the test loss is far from the optimal noise variance before this threshold, it then becomes close to it afterwards.

Yet, this transition from interpolation to generalization is slower in the 5 ReLU teacher case than in the linear one. Indeed, while interpolation does not happen anymore around $n = 2000$ in both cases, much more samples (around $n^\star = 17000$) are needed to have a simultaneously a training and testing loss close to the noise variance. These experiments suggest that the behavior predicted by Theorem 2 for a linear model also applies in more complex models such as the 5 ReLU teacher, but that the transition from interpolation to generalization can happen more slowly or with more training samples depending on the setting.

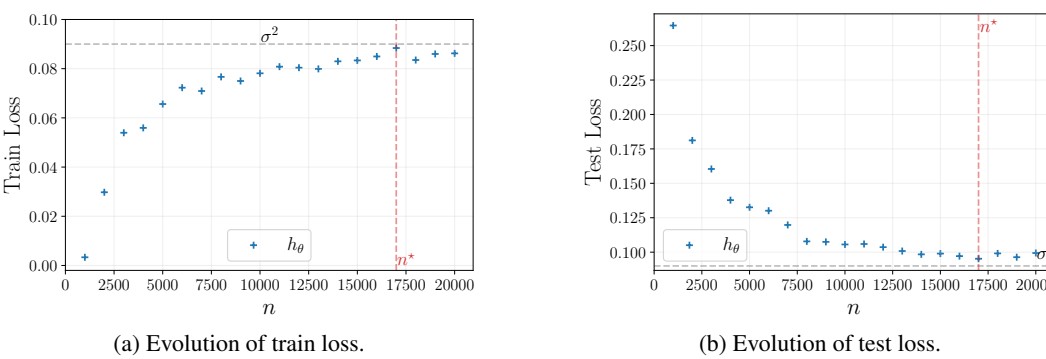

(a) Evolution of train loss.  (b) Evolution of test loss.

Figure 7: Evolution of both train and test losses at convergence with respect to the number of training samples with a 5 ReLU teacher. $\sigma^2$ corresponds to the noise variance $\mathbb{E}[\eta^2]$.

The slight difference with Figure 1 is that this optimization threshold here seems to appear for larger values of $n$.

## B  PROOF OF THEOREM 1

We recall Theorem 1 below.

**Theorem 1.** *If the marginal law of $x$ is continuous with respect to the Lebesgue measure, then for any $n \in \mathbb{N}$,*

$$\mathbb{E}_{\mathbf{X},\mathbf{y}}\left[\sup_{w\in\mathbb{S}_{d-1}}\sup_{D_n\in\mathfrak{D}_n(w,\mathbf{0})}\|D_n - D(w)\|_2\right] = \mathcal{O}\left(\sqrt{\frac{d\log n}{n}\mathbb{E}_\mu[\|yx\|_2^2]}\right),$$

*where for any $w \in \mathbb{S}_{d-1}$, $D(w) = \mathbb{E}_\mu[\mathbb{1}_{w^\top x>0}yx]$.*

*Proof.* We first show a similar result on the following expectation

$$\mathbb{E}_{\mathbf{X},\mathbf{y}}\left[\sup_{w\in\mathbb{S}_{d-1}}\|D_n(w) - D(w)\|_2\right] = \mathcal{O}\left(\sqrt{\frac{d\log n}{n}\mathbb{E}_\mu[\|yx\|_2^2]}\right), \tag{9}$$

where we recall $D_n(w) = \frac{1}{n} \sum_{k=1}^n \mathbb{1}_{w^\top x_k > 0} y_k x_k$. We bound this expectation using typical uniform bound techniques for empirical processes.

A symmetrization argument allows to show, for i.i.d. Rademacher random variables $\varepsilon_k \in \{-1, 1\}$ (see Van Der Vaart and Wellner, 2023, Lemma 2.3.1.):

$$\mathbb{E}_{\mathbf{X},\mathbf{y}} \left[ \sup_{w \in \mathbb{S}_{d-1}} \|D_n(w) - D(w)\|_2 \right] \le 2 \, \mathbb{E}_{\mathbf{X},\mathbf{y}} \left[ \mathbb{E}_{\boldsymbol{\varepsilon}} \left[ \sup_{w \in \mathbb{S}_{d-1}} \left\| \frac{1}{n} \sum_{k=1}^n \mathbb{1}_{w^\top x_k > 0} \varepsilon_k y_k x_k \right\|_2 \mid \mathbf{X}, \mathbf{y} \right] \right]. \tag{10}$$

From there, it remains to bound for any value of $\mathbf{X}, \mathbf{y}$ the conditioned expectation $\mathbb{E}_{\boldsymbol{\varepsilon}}[\cdot \mid \mathbf{X}, \mathbf{y}]$. We consider in the following a fixed value of $\mathbf{X}, \mathbf{y}$. Note that the vector $\sum_{k=1}^n \mathbb{1}_{w^\top x_k > 0} \varepsilon_k y_k x_k$, actually only depends on $w$ in the value of the vector $(\mathbb{1}_{w^\top x_k > 0})_{k \in [n]}$. Define

$$\mathcal{A}(\mathbf{X}, \mathbf{y}) = \left\{ (\mathbb{1}_{w^\top x_k > 0})_{k \in [n]} \mid w \in \mathbb{R}^d \right\}. \tag{11}$$

We thus have the equality:

$$\sup_{w \in \mathbb{S}_{d-1}} \left\| \frac{1}{n} \sum_{k=1}^n \mathbb{1}_{w^\top x_k > 0} \varepsilon_k y_k x_k \right\|_2 = \sup_{\mathbf{u} \in \mathcal{A}(\mathbf{X}, \mathbf{y})} \left\| \frac{1}{n} \sum_{k=1}^n u_k \varepsilon_k y_k x_k \right\|_2.$$

Moreover, classical geometric arguments (see e.g. Cover, 1965, Theorem 1) allow to bound $\mathrm{card}(\mathcal{A}(\mathbf{X}, \mathbf{y}))$ for any $\mathbf{X}, \mathbf{y}$:

$$\mathrm{card}(\mathcal{A}(\mathbf{X}, \mathbf{y})) \le 2 \sum_{k=0}^{d-1} \binom{n-1}{k}$$
$$= \mathcal{O}(n^d). \tag{12}$$

From there, we will bound individually for each $\mathbf{u} \in \mathcal{A}(\mathbf{X}, \mathbf{y})$ the norm of $\frac{1}{n} \sum_{k=1}^n u_k \varepsilon_k y_k x_k$ and use a union bound argument.

Let $\mathbf{u} \in \mathcal{A}(\mathbf{X}, \mathbf{y})$. Define $Z \in \mathbb{R}^{d \times n}$ the matrix whose column $k$ is given by $Z^{(k)} = \frac{1}{n} u_k y_k x_k$. Then note that $\frac{1}{n} \sum_{k=1}^n u_k \varepsilon_k y_k x_k = Z\boldsymbol{\varepsilon}$. Hanson-Wright inequality then allows to bound the following probability (see Rudelson and Vershynin, 2013, Theorem 2.1) for some universal constant $c > 0$ and any $t \ge 0$:

$$\mathbb{P}_{\boldsymbol{\varepsilon}} \left( \left| \|Z\boldsymbol{\varepsilon}\|_2 - \|Z\|_{\mathrm{F}} \right| > t \mid \mathbf{X}, \mathbf{y} \right) \le 2e^{-\frac{ct^2}{\|Z\|_{\mathrm{op}}^2}},$$

where $\|Z\|_{\mathrm{F}}$ and $\|Z\|_{\mathrm{op}}$ respectively denote the Frobenius and operator norm of $Z$. In particular, noting that $\|Z\|_{\mathrm{op}} \le \|Z\|_{\mathrm{F}}$, this last equation implies that for any $t > 0$

$$\mathbb{P}_{\boldsymbol{\varepsilon}} \left( \|Z\boldsymbol{\varepsilon}\|_2 > (1+t)\|Z\|_{\mathrm{F}} \mid \mathbf{X}, \mathbf{y} \right) \le 2e^{-ct^2}. \tag{13}$$

Moreover, note that

$$\|Z\|_{\mathrm{F}} = \sqrt{\sum_{k=1}^n \|Z^{(k)}\|_2^2}$$
$$= \sqrt{\sum_{k=1}^n \frac{1}{n^2} \|u_k y_k x_k\|_2^2} \le \sqrt{\frac{1}{n} C(Z)},$$

where $C(Z) = \frac{1}{n} \sum_{k=1}^n \|y_k x_k\|_2^2$ does not depend on $\mathbf{u}$.

Rewriting Equation (13) with this last inequality, and with $\delta = 2e^{-ct^2}$, we finally have for each $\mathbf{u} \in \mathcal{A}(\mathbf{X}, \mathbf{y})$:

$$\mathbb{P}_{\boldsymbol{\varepsilon}} \left( \|Z\boldsymbol{\varepsilon}\|_2 > \left(1 + \sqrt{\frac{1}{c} \ln(2/\delta)}\right) \sqrt{\frac{C(Z)}{n}} \mid \mathbf{X}, \mathbf{y} \right) \le \delta.$$

Considering a union bound over all the $\mathbf{u} \in \mathcal{A}(\mathbf{X}, \mathbf{y})$, we have for some universal constant $c' > 0$, thanks to Equation (12):

$$\mathbb{P}_{\boldsymbol{\varepsilon}} \left( \exists \mathbf{u} \in \mathcal{A}(\mathbf{X}, \mathbf{y}), \|Z\boldsymbol{\varepsilon}\|_2 > \left( 1 + \sqrt{c'(\log(2/\delta) + d\log(n) + 1)} \right) \sqrt{\frac{C(Z)}{n}} \mid \mathbf{X}, \mathbf{y} \right) \le \delta. \quad (14)$$

Moreover, conditioned on $\mathbf{X}, \mathbf{y}$, $\|Z\boldsymbol{\varepsilon}\|_2$ is almost surely bounded by $\sqrt{n}\|Z\|_{\mathrm{op}}$, and so by $\sqrt{C(Z)}$. A direct bound on the expectation can then be derived using Equation (14) with $\delta = n^{-d}$:

$$\mathbb{E}_{\boldsymbol{\varepsilon}} \left[ \sup_{\mathbf{u} \in \mathcal{A}(\mathbf{X}, \mathbf{y})} \left\| \frac{1}{n} \sum_{k=1}^{n} u_k \varepsilon_k y_k x_k \right\|_2 \right] = \mathcal{O}\left( \sqrt{\frac{d\log n}{n}} + n^{-d} \right) \sqrt{C(Z)}.$$

Wrapping up with Equation (11) and Equation (10) then allows to derive Equation (9),

$$\mathbb{E}_{\mathbf{X}, \mathbf{y}} \left[ \sup_{w \in \mathbb{S}_{d-1}} \|D_n(w) - D(w)\|_2 \right] = \mathcal{O}\left( \sqrt{\frac{d\log n}{n}} \right) \mathbb{E}_{\mathbf{X}, \mathbf{y}} \left[ \sqrt{C(Z)} \right]$$

$$\le \mathcal{O}\left( \sqrt{\frac{d\log n}{n}} \right) \sqrt{\mathbb{E}_{\mathbf{X}, \mathbf{y}} [C(Z)]}$$

$$= \mathcal{O}\left( \sqrt{\frac{d\log n}{n}} \right) \sqrt{\mathbb{E}_{\mu} [\|yx\|_2^2]}.$$

Lemma 1 below then allows to conclude.

**Lemma 1.** *If the marginal law of $x$ is continuous with respect to the Lebesgue measure, then almost surely:*

$$\sup_{w \in \mathbb{S}_{d-1}} \sup_{D_n \in \mathfrak{D}_n(w)} \|D_n - D(w)\|_2 \le \sup_{w \in \mathbb{S}_{d-1}} \|D_n(w) - D(w)\|_2,$$

*where $D_n(w) = \frac{1}{n} \sum_{k=1}^{n} \mathbb{1}_{w^\top x_k > 0} y_k x_k$.*

$\square$

## B.1 PROOF OF LEMMA 1

First observe that if the marginal law of $x$ is continuous, then $D$ is continuous with respect to $w$.

Consider any $w \in \mathbb{S}_{d-1}$. We recall that the set $\mathfrak{D}_n(w)$ is defined as

$$\mathfrak{D}_n(w) = \left\{ -\frac{1}{n} \sum_{k=1}^{n} \eta_k y_k x_k \mid \forall k \in [n], \eta_k \begin{cases} \in [0,1] \text{ if } \langle w_j^t, x_k \rangle = 0 \\ = 1 \text{ if } \langle w_j^t, x_k \rangle > 0 \\ = 0 \text{ otherwise} \end{cases} \right\}.$$

If all the values $w^\top x_k$ are non-zero, then $\mathfrak{D}_n(w)$ is the singleton given by $D_n(w)$ and thus

$$\sup_{D_n \in \mathfrak{D}_n(w)} \|D_n - D(w)\|_2 = \|D_n(w) - D(w)\|_2.$$

Otherwise, if $w^\top x_k = 0$ for at least one $k$, observe that[5]

$$\mathfrak{D}_n(w) = \liminf_{\substack{\varepsilon \to 0 \\ \varepsilon > 0}} \mathrm{Conv}(\{D_n(w') \mid w' \in \mathcal{S} \text{ and } \|w - w'\|_2 \le \varepsilon\}),$$

where

$$\mathcal{S} = \{w' \in \mathbb{S}_{d-1} \mid {w'}^\top x_k \ne 0 \text{ for all } k\}.$$

In other words, for any $D_n \in \mathfrak{D}_n(w)$, $w$ can be approached arbitrarily closed by vectors $w_i \in \mathcal{S}$ such that for some convex combination $\boldsymbol{\eta}$,

$$D_n = \sum_i \eta_i D_n(w_i).$$

---

[5]This observation directly follows from the definition of the Clarke subdifferential.

From then, it comes that

$$\|D_n - D(w)\| \leq \sum_i \eta_i \|D_n(w_i) - D(w)\|$$

$$\leq \sum_i \eta_i (\|D_n(w_i) - D(w_i)\| + \|D(w_i) - D(w)\|)$$

$$\leq \sup_{w' \in \mathcal{S}} \|D_n(w') - D(w')\| + \sum_i \eta_i \|D(w_i) - D(w)\|.$$

Since $D$ is continuous and the $w_i$ can be chosen arbitrarily close to $w$, the right sum can be chosen arbitrarily close to 0.

In particular, we have shown that for any $D_n \in \mathfrak{D}_n(w)$,

$$\|D_n - D(w)\| \leq \sup_{w' \in \mathcal{S}} \|D_n(w') - D(w')\|.$$

This concludes the proof of Lemma 1. $\qquad\square$

## C  PROBABILITY TAIL BOUND VERSION OF THEOREM 1

While Theorem 1 bounds the maximal deviation of $D_n - D(w)$ in expectation, a high probability tail bound is also possible, as given by Theorem 3 below.

**Theorem 3.** *If the marginal law of $x$ is continuous with respect to the Lebesgue measure, then for any $n \in \mathbb{N}$ and $M \geq \mathbb{E}_\mu \left[ \|yx\|^2 \right]$,*

$$\mathbb{P}_{\mathbf{X},\mathbf{y}} \left[ \sup_{w \in \mathbb{S}_{d-1}} \sup_{D_n \in \mathfrak{D}_n(w)} \|D_n - D(w)\|_2 > 4 \left( 1 + \sqrt{c'(\log(2/\delta) + d\log(n) + 1)} \right) \sqrt{\frac{M}{n}} \right]$$

$$\leq \frac{4}{3}\delta + \frac{4}{3}\mathbb{P}_{\mathbf{X},\mathbf{y}} \left[ \frac{1}{n} \sum_{k=1}^n \|y_k x_k\|^2 > M \right].$$

*Proof.* The proof follows the same lines as the proof of Theorem 1 in Appendix B. In particular, we first want to bound in probability the term $\sup_{w \in \mathbb{S}_{d-1}} \|D_n(w) - D(w)\|_2$. In this goal, a probabilistic symmetrization argument (Van Der Vaart and Wellner, 2023, Lemma 2.3.7.) yields for any $t > 0$

$$\beta_n(y)\mathbb{P}_{\mathbf{X},\mathbf{y}} \left[ \sup_{w \in \mathbb{S}_{d-1}} \|D_n(w) - D(w)\|_2 > t \right] \leq 2\mathbb{P}_{\mathbf{X},\mathbf{y}} \left[ \mathbb{P}_{\boldsymbol{\varepsilon}} \left[ \sup_{w \in \mathbb{S}_{d-1}} \left\| \frac{1}{n} \sum_{k=1}^n \varepsilon_k \mathbb{1}_{w^\top x_k > 0} y_k x_k \right\| > \frac{t}{4} \mid \mathbf{X}, \mathbf{y} \right] \right],$$

$$\tag{15}$$

where $\beta_n(t) = 1 - \frac{4n}{t^2} \sup_{w,w' \in \mathbb{S}_{d-1}} \mathrm{Var}_\mu(\mathbb{1}_{w^\top x > 0} y w'^\top x)$. In particular here, $\beta_n(t) \geq 1 - \frac{4n}{t^2}\mathbb{E}_\mu \left[ \|yx\|^2 \right]$. Moreover, we already showed Equation (14) in the proof of Theorem 1, which states

$$\mathbb{P}_{\boldsymbol{\varepsilon}} \left( \sup_{w \in \mathbb{S}_{d-1}} \left\| \frac{1}{n} \sum_{k=1}^n \varepsilon_k \mathbb{1}_{w^\top x_k > 0} y_k x_k \right\| > \left( 1 + \sqrt{c'(\log(2/\delta) + d\log(n) + 1)} \right) \sqrt{\frac{C(\mathbf{X},\mathbf{y})}{n}} \mid \mathbf{X}, \mathbf{y} \right) \leq \delta,$$

where $C(\mathbf{X},\mathbf{y}) = \frac{1}{n} \sum_{k=1}^n \|y_k x_k\|^2$.

Equation (15) then rewrites for any $M > 0$:

$$\mathbb{P}_{\mathbf{X},\mathbf{y}} \left[ \sup_{w \in \mathbb{S}_{d-1}} \|D_n(w) - D(w)\|_2 > 4 \left( 1 + \sqrt{c'(\log(2/\delta) + d\log(n) + 1)} \right) \sqrt{\frac{M}{n}} \right]$$

$$\leq \beta_n(t)^{-1} \left( \delta + \mathbb{P}_{\mathbf{X},\mathbf{y}} \left[ \frac{1}{n} \sum_{k=1}^n \|y_k x_k\|^2 > M \right] \right),$$

with

$$t = 4 \left( 1 + \sqrt{c'(\log(2/\delta) + d\log(n) + 1)} \right) \sqrt{\frac{M}{n}}.$$

Note that for any $M \geq \mathbb{E}_\mu\left[\|yx\|^2\right]$, $\beta_n(t) \geq \frac{3}{4}$, which implies

$$\mathbb{P}_{\mathbf{X},\mathbf{y}}\left[ \sup_{w \in \mathbb{S}_{d-1}} \|D_n(w) - D(w)\|_2 > 4\left(1 + \sqrt{c'(\log(2/\delta) + d\log(n) + 1)}\right)\sqrt{\frac{M}{n}} \right]$$

$$\leq \frac{4}{3}\delta + \frac{4}{3}\mathbb{P}_{\mathbf{X},\mathbf{y}}\left[ \frac{1}{n}\sum_{k=1}^{n} \|y_k x_k\|^2 > M \right].$$

Theorem 3 then follows, thanks to Lemma 1. □

Corollary 1 below provides a simpler tail bound, directly applying Lemma 1 with Chebyshev's inequality to bound $\mathbb{P}_{\mathbf{X},\mathbf{y}}\left[\frac{1}{n}\sum_{k=1}^{n}\|y_k x_k\|^2 > M\right]$. Stronger tail bounds can be provided with specific conditions on the random variables $x_k$ and $y_k$, but the one of Corollary 1 is enough for our use in Section 4.

**Corollary 1.** *Assume the marginal law of $x$ is continuous with respect to the Lebesgue measure. Moreover, assume $\|xy\|$ admits a fourth moment. Then*

$$\mathbb{P}_{\mathbf{X},\mathbf{y}}\left[ \sup_{w \in \mathbb{S}_{d-1}} \sup_{D_n \in \mathfrak{D}_n(w)} \|D_n - D(w)\|_2 > 4\left(1 + \sqrt{c'(\log(2/\delta) + d\log(n) + 1)}\right)\sqrt{\frac{2\mathbb{E}[\|yx\|^2]}{n}} \right]$$

$$\leq \frac{4}{3}\delta + \frac{4}{3}\frac{\mathbb{E}_\mu[\|yx\|^4]}{n\mathbb{E}_\mu[\|yx\|^2]^2}.$$

*Proof.* This is a direct consequence of Theorem 3, using Chebyshev's inequality to bound $\mathbb{P}_{\mathbf{X},\mathbf{y}}\left[\frac{1}{n}\sum_{k=1}^{n}\|y_k x_k\|^2 > M\right]$. □

## D PROOF OF PROPOSITION 1

In the following proof, we define the following subsets of the unit sphere in dimension $d$ for any $\delta > 0$:

$$\mathcal{H} = \{w \in \mathbb{S}_{d-1} \mid D(w) \text{ satisfies Equation (6)}\},$$

$$\mathcal{H}(\delta) = D^{-1}\left( \bigcup_{w \in \mathcal{H}} B(D(w), \delta) \right) \cap \mathbb{S}_{d-1},$$

$$\Delta(\delta) = \min(1, \inf_{w \in \mathbb{S}_{d-1}\backslash\mathcal{H}(\delta)} \min\left( \|\frac{D(w)}{\|D(w)\|} - w\|, \|\frac{D(w)}{\|D(w)\|} + w\|, \|D(w)\| \right)).$$

Here, $B(D(w), \delta)$ denotes the open ball of radius $\delta$, centered in $D(w)$.

*Proof.* Since the marginal distribution of $X$ is continuous, the function $D : w \mapsto D(w)$ is continuous. In particular for any $\delta > 0$, the infimum defining $\Delta(\delta)$ is reached, so that $\Delta(\delta) > 0$ by definition of $\mathcal{H}$. In the following, we let $\delta = \frac{\varepsilon}{2}$. Thanks to Corollary 1, with probability at least $1 - \mathcal{O}_\mu\left(\frac{1}{n}\right)$,

$$\sup_{w \in \mathbb{S}_{d-1}} \sup_{D_n \in \mathfrak{D}_n(w)} \|D_n - D(w)\|_2 = \mathcal{O}_\mu\left( \sqrt{\frac{d\log n}{n}} \right).$$

In particular, we can choose $n^\star(\varepsilon) = \mathcal{O}_\mu\left( \frac{d\log\left(\frac{d}{\min(\Delta(\frac{\varepsilon}{2})^4, \varepsilon^2)}\right)}{\min}(\Delta(\frac{\varepsilon}{2})^2, \varepsilon) \right)$ large enough so that

for any $n \geq n^\star(\varepsilon)$, with probability at least $1 - \mathcal{O}_\mu\left(\frac{1}{n}\right)$:

$$\sup_{w \in \mathbb{S}_{d-1}} \sup_{D_n \in \mathfrak{D}_n(w)} \|D_n - D(w)\|_2 \leq \frac{1}{2}\min\left( \Delta(\frac{\varepsilon}{2})^2, \varepsilon \right). \tag{16}$$

We assume in the following of the proof that Equation (16) holds.

Consider an extremal vector $D_n$ of the finite data $(\mathbf{X}, \mathbf{y})$. By definition, there is some $w \in \mathbb{S}_{d-1}$ such that $D \in \mathcal{D}_n(w)$ and either

1. $D_n = \mathbf{0}$,

2. or $A_n(D_n) = A_n(w)$,

3. or $A_n(D_n) = -A_n(w)$.

In the first case, Equation (16) yields that $\|D(w)\|_2 < \Delta(\frac{\varepsilon}{2})$. Necessarily, by definition of $\Delta(\frac{\varepsilon}{2})$, $w \in \mathcal{H}(\frac{\varepsilon}{2})$. This means by definition of $\mathcal{H}(\frac{\varepsilon}{2})$ that there exists $D^\star \in \mathbb{R}^d$ satisfying Equation (6), such that

$$\|D(w) - D^\star\|_2 \leq \frac{\varepsilon}{2}.$$

In particular, using Equation (16) again yields $\|D_n - D^\star\|_2 \leq \varepsilon$.

In the second case $(A_n(D_n) = A_n(w))$, we can assume $D_n \neq 0$. In that case, as $\frac{D_n}{\|D_n\|}$ have the same activations, $\mathcal{D}_n(w) = \mathcal{D}_n(\frac{D_n}{\|D_n\|})$, i.e., we can assume without loss of generality that $w = \frac{D_n}{\|D_n\|}$ here. Similarly to the first case, if $w \in \mathcal{H}(\frac{\varepsilon}{2})$, then there exists $D^\star \in \mathbb{R}^d$ satisfying Equation (6), such that $\|D_n - D^\star\|_2 \leq \varepsilon$.

Let us show by contradiction that indeed $w \in \mathcal{H}(\frac{\varepsilon}{2})$. Assume $w \notin \mathcal{H}(\frac{\varepsilon}{2})$. In particular, $\|D(w)\|_2 \geq \Delta(\frac{\varepsilon}{2})$. We can now bound the norm of $\frac{D(w)}{\|D(w)\|} - w$:

$$\left\| \frac{D(w)}{\|D(w)\|} - w \right\|_2 = \left\| \frac{D(w)}{\|D(w)\|} - \frac{D_n}{\|D_n\|} \right\|_2$$

$$= \left\| \frac{D(w) - D_n}{\|D(w)\|} + \frac{D_n}{\|D_n\|} \left( \frac{\|D_n\| - \|D(w)\|}{\|D(w)\|} \right) \right\|_2$$

$$\leq \frac{\|D(w) - D_n\|}{\|D(w)\|} + \frac{\big| \|D_n\| - \|D(w)\| \big|}{\|D(w)\|}$$

$$\leq 2 \frac{\|D(w) - D_n\|}{\|D(w)\|}$$

$$\leq 2 \frac{\Delta(\frac{\varepsilon}{2})^2}{2\Delta(\frac{\varepsilon}{2})} = \Delta(\frac{\varepsilon}{2}).$$

By definition of $\Delta(\frac{\varepsilon}{2})$, this actually implies that $w \in \mathcal{H}(\frac{\varepsilon}{2})$, which contradicts the initial assumption. We thus indeed have $w \in \mathcal{H}(\frac{\varepsilon}{2})$, leading to the existence of a $D^\star$ with the wanted properties such that $\|D_n - D^\star\|_2 \leq \varepsilon$. In the third case $(A_n(D_n) = -A_n(w))$, symmetric arguments lead to the same conclusion, which concludes the proof of Proposition 1. $\qquad\square$

# E  PROOF OF THEOREM 2

## E.1  NOTATIONS AND FIRST CLASSICAL RESULTS

In the whole Appendix E, we define $\Sigma = \mathbb{E}[x x^\top]$ and

$$\Sigma_{n,+} = \frac{1}{n} \sum_{k \in \mathcal{S}_+} x_k x_k^\top \quad , \quad \Sigma_{n,-} = \frac{1}{n} \sum_{k \in \mathcal{S}_-} x_k x_k^\top$$

and the following set of neurons:

$$\mathcal{I}_+ = \{i \in [m] \mid a_i(0) \geq 0\} \quad \text{and} \quad \mathcal{I}_- = \{i \in [m] \mid a_i(0) < 0\}.$$

We first start by stating the following, known balancedness lemma (see, e.g., Arora et al., 2019; Boursier et al., 2022).

**Lemma 2.** *For any $i \in [m]$ and $t \in \mathbb{R}_+$, $a_i(t)^2 - \|w_i(t)\|^2 = a_i(0)^2 - \|w_i(0)\|^2$.*

Lemma 2 can be simply proved by a direct computation of the derivative of $a_i(t)^2 - \|w_i(t)\|^2$. Thanks to Equation (8), this yields that the sign $a_i(t)$ is constant over time, and thus partitioned by the sets of neurons $\mathcal{I}_+$ and $\mathcal{I}_-$.

Also, note that with probability $1 - \frac{1}{2^{m-1}}$, the sets $\mathcal{I}_+$ and $\mathcal{I}_-$ are both non empty, which is assumed to hold in the following of the section.

In this section, all the $\mathcal{O}, \Theta$ and $\Omega$ notations hide constants depending on the fourth moment of $\eta$, the norm of $\beta^\star$ and the constant $c$ of Assumption 1. Note that due to the sub-Gaussian property of $x$, its $k$-th moment can be bounded as $\mathbb{E}[\|x\|^k] \mathcal{O}\left(d^{\frac{k}{2}}\right)$ for any $k$.

### E.2 Phase 1: early alignment

**Lemma 3.** *If Assumption 1 holds, there exists $\lambda^\star = \Theta(\frac{1}{d})$ and $n^\star = \Theta(d^3 \log d)$ such that for any $\lambda \leq \lambda^\star$ and $\varepsilon \in (0, \frac{1}{4})$, $n \geq n^\star$ and for $\tau = \frac{\varepsilon \ln(1/\lambda)}{\|\Sigma\beta^\star\|}$, with probability $1 - \mathcal{O}(\frac{1}{n})$:*

*1. output weights do not change until $\tau$:*
$$\forall t \leq \tau, \forall j \in [m], |a_j(0)|\lambda^{2\varepsilon} \leq |a_j(t)| \leq |a_j(0)|\lambda^{-2\varepsilon};$$

*2. all neurons align with $\pm\Sigma\beta^\star$:*
$$\forall i \in [m], \quad \langle \frac{w_i(\tau)}{a_i(\tau)}, \Sigma\beta^\star \rangle = \|\Sigma\beta^\star\| - \mathcal{O}\left(\lambda^\varepsilon + \sqrt{\frac{d^2 \log n}{n}}\right).$$

*Proof.* We start the proof by computing $D(w)$ for any $w \in \mathbb{S}_{d-1}$:
$$\begin{aligned}
D(w) &= \mathbb{E}[\mathbb{1}_{w^\top x > 0} y x] \\
&= \mathbb{E}[\mathbb{1}_{w^\top x > 0} x x^\top \beta^\star] \\
&= \frac{1}{2} \left(\mathbb{E}[\mathbb{1}_{w^\top x > 0} x x^\top] + \mathbb{E}[\mathbb{1}_{w^\top x < 0} x x^\top]\right) \beta^\star \\
&= \frac{\Sigma\beta^\star}{2}.
\end{aligned}$$

The second inequality comes from the independence between $x$ and $\eta$, the third one comes from the symmetry of the distribution of $x$ and the last one by continuity of this distribution.

Corollary 1 additionally implies that for some $n^\star = \Theta(d^3 \log d)$ and any $n \geq n^\star$, with probability at least $1 - \mathcal{O}(\frac{1}{n})$,

$$\sup_{w \in \mathbb{S}_{d-1}} \sup_{D_n \in \mathfrak{D}_n(w)} \|D_n - D(w)\|_2 \leq \mathcal{O}\left(\sqrt{\frac{d^2 \log n}{n}}\right) \leq \frac{\alpha}{\sqrt{d}}, \tag{17}$$

with $\alpha = \frac{1}{4} \min(c - \sqrt{d}\|\Sigma\beta^\star - \beta^\star\|, \|\Sigma\beta^\star\|)$. Note[6] that $\alpha > 0$ thanks to the fourth point of Assumption 1. Moreover, using typical concentration inequality for sub-Gaussian vectors, we also have with probability $1 - \mathcal{O}(\frac{1}{n})$:

$$\frac{\sum_{k=1}^n \|x_k\|^2}{n} \leq 2\mathbb{E}_{\mu_X}[\|x\|^2] = \mathcal{O}(d). \tag{18}$$

We assume in the following of the proof that both Equations (17) and (18) hold.

---

[6]The additional $d$ dependence comes from the expectation of $\|yx\|^2$ in the square root. Additionally, the probability bound comes from the fact that $\frac{\mathbb{E}_\mu[\|yx\|^4]}{\mathbb{E}_\mu[\|yx\|^2]^2} = \mathcal{O}(1)$ here.

Since $D(w) = \frac{\Sigma\beta^\star}{2}$ for any $w$, we have for any $w \in \mathbb{S}_{d-1}$, $D_n \in \mathfrak{D}_n(w)$ and $k \in \mathcal{S}_+$:

$$x_k^\top D_n = x_k^\top (D_n - D(w)) + \frac{1}{2} x_k^\top (\Sigma\beta^\star - \beta^\star) + \frac{1}{2} x_k^\top \beta^\star$$

$$\geq \|x_k\| \left( -\|D_n - D(w)\| - \frac{1}{2}\|\Sigma\beta^\star - \beta^\star\| + \frac{c}{2\sqrt{d}} \right)$$

$$\geq \frac{\alpha}{\sqrt{d}}\|x_k\| > 0.$$

Similarly for any $k \in \mathcal{S}_-$, $x_k^\top D_n < 0$. This directly implies here that there are only two extremal vectors here:

$$D_n(\beta^\star) = \Sigma_{n,+}\beta^\star + \frac{1}{n}\sum_{k \in \mathcal{S}_+} \eta_k x_k,$$

$$D_n(-\beta^\star) = \Sigma_{n,-}\beta^\star + \frac{1}{n}\sum_{k \in \mathcal{S}_-} \eta_k x_k. \tag{19}$$

We can now show, similarly to Boursier and Flammarion (2024), the early alignment phenomenon in the first phase.[7]

1. First note that Equation (17) and the definition of $\alpha$ imply that for any $w$:

$$\|D_n(w)\| \leq \|\Sigma\beta^\star\|. \tag{20}$$

We define $t_1 = \min\{t \geq 0 \mid \sum_{j=1}^m a_j(t)^2 \geq \lambda^{2-4\varepsilon}\}$.

For any $i \in [m]$ and $t \in [0, t_1]$, Equation (4) rewrites:

$$\left| \frac{da_i(t)}{dt} \right| = \left| w_i(t)^\top D_n^i(t) \right|$$

$$\leq |a_i(t)| \left( \max_{w \in \mathbb{S}_{d-1}} \|D_n(w)\| + \frac{\sum_{k=1}^n \|x_k\|^2}{n} \lambda^{2-4\varepsilon} \right)$$

$$\leq |a_i(t)| \left( \|\Sigma\beta^\star\| + 2\mathbb{E}[\|x\|^2]\lambda^{2-4\varepsilon} \right).$$

As a consequence, a simple Grönwall argument yields that for any $t \in [0, t_1]$:

$$|a_i(t)| \leq |a_i(0)| \exp(t\|\Sigma\beta^\star\| + 2t\mathbb{E}[\|x\|^2]\lambda^{2-4\varepsilon}).$$

In particular, for our choice of $\tau$, for a small enough $\lambda^\star = \mathcal{O}\left(d^{-\frac{1}{2-4\varepsilon}}\right)$, for any $t \leq \min(\tau, t_1)$:

$$|a_i(t)| < |a_i(0)|\lambda^{-\varepsilon}. \tag{21}$$

Note that this implies that $t < t_1$, i.e., $\tau < t_1$. As a consequence. Moreover, we can also show that $|a_i(t)| > |a_i(0)|\lambda^\varepsilon$ for any $t \leq \tau$, which implies the first point of Lemma 3.

2. For the second point, let $i \in \mathcal{I}_+$ and denote $\overline{w}_i(t) = \frac{w_i(t)}{a_i(t)}$. Thanks to Lemma 2, $\overline{w}_i(t) \in B(0, 1)$ and $a_i(t)$ is of constant sign. Also, for almost any $t \in [0, \tau]$:

$$\frac{d\overline{w}_i(t)}{dt} \in \mathcal{D}_n(w_i(t), \theta(t)) - \langle \overline{w}_i(t), \mathcal{D}_n(w_i(t), \theta(t)) \rangle \overline{w}_i(t).$$

---

[7]We could directly reuse Theorem 1 from Boursier and Flammarion (2024) here, but it would not allow us to choose an initialisation scale $\lambda^\star$ that does not depend on $n$.

Since $a_i(t) > 0$ for $i \in \mathcal{I}_+$,

$$
\begin{aligned}
\frac{\mathrm{d}\langle \overline{w}_i(t), \Sigma\beta^\star \rangle}{\mathrm{d}t} &\in \langle \mathcal{D}_n(w_i(t), \theta(t)), \Sigma\beta^\star \rangle - \langle \overline{w}_i(t), \mathcal{D}_n(w_i(t), \theta(t)) \rangle \langle \overline{w}_i(t), \Sigma\beta^\star \rangle \\
&\geq \inf_{D_n \in \mathfrak{D}_n(w_i(t), \theta(t))} \langle D_n, \Sigma\beta^\star \rangle - \langle \overline{w}_i(t), D_n \rangle \langle \overline{w}_i(t), \Sigma\beta^\star \rangle \\
&\geq \inf_{D_n \in \mathfrak{D}_n(w_i(t))} \langle D_n, \Sigma\beta^\star \rangle - \langle \overline{w}_i(t), D_n \rangle \langle \overline{w}_i(t), \Sigma\beta^\star \rangle - 2\|\Sigma\beta^\star\|\lambda^{2-4\varepsilon} \\
&\geq \langle D(w_i(t)), \Sigma\beta^\star \rangle - \langle \overline{w}_i(t), D(w_i(t)) \rangle \langle \overline{w}_i(t), \Sigma\beta^\star \rangle \\
&\quad - 2\|\Sigma\beta^\star\| \left( \lambda^{2-4\varepsilon} + \sup_{D_n \in \mathcal{D}_n(w_i(t)} \|D_n - D(w_i(t))\| \right) \\
&\geq \frac{1}{2} \left( \|\Sigma\beta^\star\|^2 - \langle \overline{w}_i(t), \Sigma\beta^\star \rangle^2 \right) - \mathcal{O}\left( \lambda^{2-4\varepsilon} + \sqrt{\frac{d^2 \log n}{n}} \right).
\end{aligned}
$$

Solutions of the ODE $f'(t) = a^2 - f(t)^2$ with $f(0) \in (-a, a)$ are of the form $f(t) = a \tanh(a(t + t_0))$ for some $t_0 \in \mathbb{R}$. By Grönwall comparison, we thus have

$$
\langle \overline{w}_i(t), \Sigma\beta^\star \rangle \geq a \tanh(\frac{a}{2}(t + t_j)), \tag{22}
$$

$$
\text{where} \quad a = \|\Sigma\beta^\star\| - \mathcal{O}\left( \lambda^{2-4\varepsilon} + \sqrt{\frac{d^2 \log n}{n}} \right)
$$

$$
\text{and} \quad \langle \overline{w}_i(0), \Sigma\beta^\star \rangle = a \tanh(\frac{a}{2} t_j).
$$

Thanks to the choice of initialisation given by Equation (8), $\|\overline{w}_i(0)\| \leq \frac{1}{2}$ and so $\langle \overline{w}_i(0), \Sigma\beta^\star \rangle \geq -\frac{1}{2}\|\Sigma\beta^\star\|_2$. Moreover, $\tanh(x) \leq -1 + 2e^{2x}$, so that

$$
-\frac{1}{2}\|\Sigma\beta^\star\| \leq a(-1 + 2e^{at_j}).
$$

Since $a = \|\Sigma\beta^\star\| - \mathcal{O}\left( \lambda^{2-4\varepsilon} + \sqrt{\frac{d^2 \log n}{n}} \right)$, this yields

$$
2a e^{at_j} \geq \frac{1}{2}\|\Sigma\beta^\star\| + \mathcal{O}\left( \lambda^{2-4\varepsilon} + \sqrt{\frac{d^2 \log n}{n}} \right).
$$

The previous inequality can be rewritten as

$$
\begin{aligned}
-2a e^{-t_j} &\geq \frac{-4a^2}{\frac{1}{2}\|\Sigma\beta^\star\| + \mathcal{O}\left( \lambda^{2-4\varepsilon} + \sqrt{\frac{d^2 \log n}{n}} \right)} \\
&\geq \frac{-8a^2}{\|\Sigma\beta^\star\|} (1 + \mathcal{O}\left( \lambda^{2-4\varepsilon} + \sqrt{\frac{d^2 \log n}{n}} \right)) \\
&\geq 8\|\Sigma\beta^\star\| + \mathcal{O}\left( \lambda^{2-4\varepsilon} + \sqrt{\frac{d^2 \log n}{n}} \right).
\end{aligned}
$$

Using that $\tanh(x) \geq 1 - 2e^{-2x}$, Equation (22) becomes at time $\tau$ and $\frac{n}{\log(n)} = \Omega(d^2)$,

$$\langle \overline{w}_i(\tau), \Sigma\beta^\star \rangle \geq a - 2ae^{-at_j}e^{-a\tau}$$

$$\geq \|\Sigma\beta^\star\| - \|\Sigma\beta^\star\|(8\|\Sigma\beta^\star\| + \mathcal{O}\left(\lambda^{2-4\varepsilon} + \sqrt{\frac{d^2\log n}{n}}\right))e^{\frac{a\varepsilon\log\lambda}{\|\Sigma\beta^\star\|}}$$

$$- \mathcal{O}\left(\lambda^{2-4\varepsilon} + \sqrt{\frac{d^2\log n}{n}}\right)$$

$$\geq \|\Sigma\beta^\star\| - \mathcal{O}\left(\lambda^\varepsilon + \sqrt{\frac{d^2\log n}{n}}\right).$$

3. The same arguments can be done with negative neurons. □

### E.3 Decoupled autonomous systems

In the remaining of the proof, we will focus on an alternative solution $(\mathsf{w}, \mathsf{a})$, which is solution of the following differential equations for any $t \geq \tau$

$$\frac{d\mathsf{w}_i(t)}{dt} = \mathsf{a}_i(t)D_+(t) \quad \text{and} \quad \frac{d\mathsf{a}_i(t)}{dt} = \langle \mathsf{w}_i(t), D_+(t)\rangle \quad \text{for any } i \in \mathcal{I}_+,$$
$$\frac{d\mathsf{w}_i(t)}{dt} = \mathsf{a}_i(t)D_-(t) \quad \text{and} \quad \frac{d\mathsf{a}_i(t)}{dt} = \langle \mathsf{w}_i(t), D_-(t)\rangle \quad \text{for any } i \in \mathcal{I}_-, \tag{23}$$

where

$$D_+(t) = \frac{1}{n}\sum_{k\in\mathcal{S}_+}\left(\sum_{i\in\mathcal{I}_+}\mathsf{a}_i(t)\langle\mathsf{w}_i(t), x_k\rangle - y_k\right)x_k,$$

$$D_-(t) = \frac{1}{n}\sum_{k\in\mathcal{S}_-}\left(\sum_{i\in\mathcal{I}_-}\mathsf{a}_i(t)\langle\mathsf{w}_i(t), x_k\rangle - y_k\right)x_k$$

and with the initial condition $\mathsf{w}_i(\tau), \mathsf{a}_i(\tau) = w_i(\tau), a_i(\tau)$ for any $i \in [m]$. We also note in the following $\overline{\mathsf{w}}_i = \frac{\mathsf{w}_i}{\mathsf{a}_i}$ and the estimations of the training data $x_k$ for any $k \in [n]I$ as:

$$h_\vartheta(x_k) = \begin{cases} \sum_{i\in\mathcal{I}_+}\mathsf{a}_i\langle\mathsf{w}_i, x_k\rangle & \text{if } k \in \mathcal{S}_+ \\ \sum_{i\in\mathcal{I}_-}\mathsf{a}_i\langle\mathsf{w}_i, x_k\rangle & \text{if } k \in \mathcal{S}_- \end{cases}.$$

This construction allows to study separately the dynamics of both sets of neurons $\mathcal{I}_+$ and $\mathcal{I}_-$, without any interaction between each other. As precised by Lemma 4 below, $\mathsf{w}_i, \mathsf{a}_i$ coincide with $w_i, a_i$ as long as the neurons all remain in the sector they are at the end of the early alignment phase.

**Lemma 4.** *Define* $T_+ = \inf\{t \geq \tau \mid \exists(i, k) \in \mathcal{I}_+ \times [n], \operatorname{sign}(x_k^\top \mathsf{w}_i(t)) \neq \operatorname{sign}(x_k^\top \beta^\star)\}$ *and* $T_- = \inf\{t \geq \tau \mid \exists(i, k) \in \mathcal{I}_- \times [n], \operatorname{sign}(x_k^\top \mathsf{w}_i(t)) \neq -\operatorname{sign}(x_k^\top \beta^\star)\}$.

*Then for any* $i \in [m]$ *and any* $t \in [\tau, \min(T_+, T_-)]$: $(\mathsf{w}_i(t), \mathsf{a}_i(t)) = (w_i(t, a_i(t))$. *Moreover, for any* $t \in [\tau, \min(T_+, T_-)]$ *and* $k \in [n]$, $h_{\vartheta(t)}(x_k) = h_{\theta(t)}(x_k)$.

While analysing the complete dynamics of $(\mathsf{w}, \mathsf{a})$, we will see that both $T_+$ and $T_-$ are infinite in the considered range of parameters, thus leading to a complete description of the dynamics of $(w, a)$.

*Proof.* Thanks to the definition of $T_+$ and $T_-$, the evolution of $(w_i(t), a_i(t))$ given by 4 coincides with the evolution of $(\mathsf{w}_i(t), \mathsf{a}_i(t))$ given by Equation (23) for $t \in [\tau, \min(T_+, T_-)]$. The associated ODE is Lipschitz on the considered time interval and thus admits a unique solution, hence leading to $(\mathsf{w}_i(t), \mathsf{a}_i(t)) = (w_i(t, a_i(t))$ on the considered interval. The equality $h_{\vartheta(t)}(x_k) = h_{\theta(t)}(x_k)$ directly derives from the ReLU activations and definitions of $T_+$ and $T_-$. □

### E.4 PHASE 2: NEURONS SLOW GROWTH

For some $\varepsilon_2 > 0$, we define the following stopping time for any $\circ \in \{+, -\}$:

$$\tau_{2,\circ} = \inf\{t \geq \tau \mid \sum_{i \in \mathcal{I}_\circ} \mathsf{a}_i(t)^2 \geq \varepsilon_2\}.$$

**Lemma 5.** *If Assumption 1 holds, for any $\varepsilon \in (0, \frac{1}{4})$, there exist $\lambda^\star = \Theta(\frac{1}{d})$, $\varepsilon_2^\star = \Theta(d^{-\frac{3}{2}})$ and $n^\star = \Theta(d^3 \log d)$ such that for any $\lambda \leq \lambda^\star$, $n \geq n^\star$, $\circ \in \{+, -\}$, $\varepsilon_2 \in [\lambda^{2-4\varepsilon}, \varepsilon_2^\star]$, with probability $1 - \mathcal{O}(\frac{1}{n} + \frac{1}{2^m})$, $\tau_{2,\circ} < +\infty$ and at this time,*

    *1. neurons in $\mathcal{I}_\circ$ are aligned with each other*

$$\forall i, j \in \mathcal{I}_\circ, \quad \langle \overline{\mathsf{w}}_j(\tau_{2,\circ}), \overline{\mathsf{w}}_i(\tau_{2,\circ}) \rangle = 1 - \mathcal{O}\left(\frac{\lambda^{\frac{1}{2}}}{\varepsilon_2}\right);$$

    *2. neurons in $\mathcal{I}_\circ$ are in the same cone as $\circ\beta^\star$ for any $t \in [\tau, \tau_{2,\circ}]$:*

$$\forall i \in \mathcal{I}_\circ, \quad \min_{k \in \mathcal{S}_\circ} \langle \overline{\mathsf{w}}_i(\tau_{2,\circ}), \frac{x_k}{\|x_k\|} \rangle = \Omega(\frac{1}{\sqrt{d}}) \quad and \quad \max_{k \in \mathcal{S}_{-\circ}} \langle \overline{\mathsf{w}}_i(\tau_{2,\circ\circ}), \frac{x_k}{\|x_k\|} \rangle = -\Omega(\frac{1}{\sqrt{d}}).$$

*Proof.* In the following, we assume without loss of generality that $\circ = +$. Additionally, we assume that the random event $\mathcal{I}_+ \neq \emptyset$ and Equations (17) and (18) hold. First, by definition of $\tau_{2,+}$, for any $t \in [\tau, \tau_{2,+}]$:

$$\|D_+(t) - D_n(\beta^\star)\|_2 \leq \frac{1}{n} \left( \sum_{i \in \mathcal{I}_+} \mathsf{a}_i(t)^2 \right) \sum_{k \in \mathcal{S}_+} \|x_k\|^2$$

$$\leq 2\varepsilon_2 \mathbb{E}_{\mu_X}[\|x\|^2].$$

This also implies with Equation (20) that $\|D_+(t)\| \leq \|\Sigma\beta^\star\| + 2\varepsilon_2 \mathbb{E}_{\mu_X}[\|x\|^2]$. Additionally, we have with Equation (17) that

$$\|D_+(t) - \frac{\Sigma\beta^\star}{2}\|_2 \leq \|D_+(t) - D_n(\beta^\star)\|_2 + \|D_n(\beta^\star) - D(\beta^\star)\|_2$$

$$\leq \mathcal{O}\left(d\varepsilon_2 + \sqrt{\frac{d^2 \log n}{n}}\right). \tag{24}$$

Then for any $k \in \mathcal{S}_+, i \in \mathcal{I}_+$ and $t \in [\tau, \tau_{2,+}]$, as long as $\langle \overline{\mathsf{w}}_i(t), x_k \rangle \geq 0$,

$$\frac{\mathrm{d}\langle \overline{\mathsf{w}}_i(t), \frac{x_k}{\|x_k\|} \rangle}{\mathrm{d}t} = \langle D_+(t), \frac{x_k}{\|x_k\|} \rangle - \langle D_+(t), \overline{\mathsf{w}}_i(t) \rangle \langle \overline{\mathsf{w}}_i(t), \frac{x_k}{\|x_k\|} \rangle$$

$$\geq \langle D_n(\beta^\star), \frac{x_k}{\|x_k\|} \rangle - 2\varepsilon_2 \mathbb{E}_{\mu_X}[\|x\|^2] - \|D_+(t)\| \langle \overline{\mathsf{w}}_i(t), \frac{x_k}{\|x_k\|} \rangle$$

$$\geq \langle D_n(\beta^\star), \frac{x_k}{\|x_k\|} \rangle - 2\varepsilon_2 \mathbb{E}_{\mu_X}[\|x\|^2] - (\|D_n(\beta^\star)\| + 2\varepsilon_2 \mathbb{E}_{\mu_X}[\|x\|^2]) \langle \overline{\mathsf{w}}_i(t), \frac{x_k}{\|x_k\|} \rangle. \tag{25}$$

As $\langle D_n(\beta^\star), \frac{x_k}{\|x_k\|} \rangle \geq \frac{\alpha}{\sqrt{d}}$ (Equation 19), thanks to Lemma 3 and the third point in Assumption 1, and $\|D_n(\beta^\star)\| = \mathcal{O}(1)$, for a small enough $\varepsilon_2^\star = \Theta(d^{-\frac{3}{2}})$, $\min_{k \in \mathcal{S}_+} \langle \overline{\mathsf{w}}_i(\tau), \frac{x_k}{\|x_k\|} \rangle = \Omega(\frac{1}{\sqrt{d}})$. Equation (27) then implies for a small enough choice of $\varepsilon_2^\star = \Theta(d^{-\frac{3}{2}})$ and $\varepsilon_2 \leq \varepsilon_2^\star$:

$$\min_{t \in [\tau, \tau_{2,+}]} \min_{k \in \mathcal{S}_+} \langle \overline{\mathsf{w}}_i(\tau), \frac{x_k}{\|x_k\|} \rangle = \Omega(\frac{1}{\sqrt{d}}). \tag{26}$$

Similarly, we can also show

$$\max_{t \in [\tau, \tau_{2,+}]} \max_{k \in \mathcal{S}_-} \langle \overline{\mathsf{w}}_i(\tau), \frac{x_k}{\|x_k\|} \rangle = -\Omega(\frac{1}{\sqrt{d}}),$$

which implies the second point of Lemma 5. Actually, we even have for this choice of parameters the more precise inequality (for the same reasons) that for any $k \in \mathcal{S}_+, i \in \mathcal{I}_+$ and $t \in [\tau, \tau_{2,+}]$,

$$\langle \overline{\mathsf{w}}_i(\tau), \frac{x_k}{\|x_k\|} \rangle \geq \langle \frac{D_n(\beta^\star)}{\|D_n(\beta^\star)\|}, \frac{x_k}{\|x_k\|} \rangle - \mathcal{O}(d\varepsilon_2). \tag{27}$$

We now simultaneously lower and upper bound the duration of the second phase $\tau_{2,+} - \tau_2$. For any $t \in [\tau, \tau_{2,+}]$:

$$\frac{1}{2}\frac{\mathrm{d}\sum_{i\in\mathcal{I}_+}\mathsf{a}_i(t)^2}{\mathrm{d}t} = \sum_{i\in\mathcal{I}_+}\mathsf{a}_i(t)^2\langle\overline{\mathsf{w}}_i(t), D_+(t)\rangle$$

$$= \frac{1}{n}\sum_{i\in\mathcal{I}_+}\mathsf{a}_i(t)^2\sum_{k\in\mathcal{S}_+}(y_k - h_{\vartheta(t)}(x_k))\langle\overline{\mathsf{w}}_i(t), x_k\rangle \tag{28}$$

$$\geq \sum_{i\in\mathcal{I}_+}\mathsf{a}_i(t)^2\left(\frac{\sum_{k\in\mathcal{S}_+}y_k\|x_k\|\langle\overline{\mathsf{w}}_i(t), \frac{x_k}{\|x_k\|}\rangle}{n} - \frac{\varepsilon_2\sum_{k\in\mathcal{S}_+}\|x_k\|^2}{n}\right).$$

Note that $\mathbb{E}[\mathbb{1}_{k\in\mathcal{S}_+}y_k\|x_k\|] \geq \frac{c\mathbb{E}[\|x\|^2]}{2\sqrt{d}}$. Using Chebyshev inequality, we thus have for a small enough choice of $\varepsilon_2^\star = \Theta(d^{-\frac{3}{2}})$, for any $t \in [\tau, \tau_{2,+}]$:

$$\frac{\mathrm{d}\sum_{i\in\mathcal{I}_+}\mathsf{a}_i(t)^2}{\mathrm{d}t} \geq \Omega(1)\sum_{i\in\mathcal{I}_+}\mathsf{a}_i(t)^2.$$

A Grönwall comparison then directly yields $\tau_{2,+} < \infty$.

We now want to show that the neurons $\overline{\mathsf{w}}_i$ are almost aligned at the end of the second phase. For that, we first need to lower bound the duration of the phase. Note that Equation (28), with Equation (26), also leads for any $t \in [\tau, \tau_{2,+}]$ to

$$\frac{1}{2}\frac{\mathrm{d}\sum_{i\in\mathcal{I}_+}\mathsf{a}_i(t)^2}{\mathrm{d}t} \leq \frac{1}{n}\sum_{i\in\mathcal{I}_+}\mathsf{a}_i(t)^2\sum_{k\in\mathcal{S}_+}y_k\langle\overline{\mathsf{w}}_i(t), x_k\rangle$$

$$= \sum_{i\in\mathcal{I}_+}\mathsf{a}_i(t)^2\langle\overline{\mathsf{w}}_i(t), D_n(\beta^\star)\rangle$$

$$\leq \sum_{i\in\mathcal{I}_+}\mathsf{a}_i(t)^2\|D_n(\beta^\star)\|.$$

Note that by continuity, $\sum_{i\in\mathcal{I}_+}\mathsf{a}_i(\tau_{2,+})^2 = \varepsilon_2$. As $\sum_{i\in\mathcal{I}_+}\mathsf{a}_i(\tau)^2 \leq \lambda^{2-4\varepsilon}$, thanks to Lemma 3, a Grönwall inequality argument leads to the following as $\varepsilon_2 \geq \lambda^{2-4\varepsilon}$,

$$\tau_{2,+} - \tau \geq \frac{1}{2\|D_n(\beta^\star)\|}\ln\left(\frac{\varepsilon_2}{\lambda^{2-4\varepsilon}}\right). \tag{29}$$

For any pair of neurons $i, j \in \mathcal{I}_+$, we consider the evolution of the mutual alignment:

$$\frac{\mathrm{d}\langle\overline{\mathsf{w}}_i(t), \overline{\mathsf{w}}_j(t)\rangle}{\mathrm{d}t} = \langle D_+(t), \overline{\mathsf{w}}_i(t) + \overline{\mathsf{w}}_j(t)\rangle(1 - \langle\overline{\mathsf{w}}_i(t), \overline{\mathsf{w}}_j(t)\rangle)$$

$$= (\langle D_n(\beta^\star), \overline{\mathsf{w}}_i(t) + \overline{\mathsf{w}}_j(t)\rangle - \mathcal{O}(\varepsilon_2))(1 - \langle\overline{\mathsf{w}}_i(t), \overline{\mathsf{w}}_j(t)\rangle). \tag{30}$$

Moreover, Equation (27) leads to the following alignment between $\overline{\mathsf{w}}_i(t)$ and $D_n(\beta^\star)$ for any $t \in [\tau, \tau_{2,+}]$:

$$\langle D_n(\beta^\star), \overline{\mathsf{w}}_i(t) \rangle = \frac{1}{n} \sum_{k \in \mathcal{S}_+} y_k \langle x_k, \overline{\mathsf{w}}_i(t) \rangle$$

$$= \frac{1}{n} \sum_{k \in \mathcal{S}_+} y_k \langle x_k, \frac{D_n(\beta^\star)}{\|D_n(\beta^\star)\|} \rangle - \|x_k\| \mathcal{O}(d\varepsilon_2)$$

$$= \langle D_n(\beta^\star), \frac{D_n(\beta^\star)}{\|D_n(\beta^\star)\|} \rangle - \mathcal{O}(d\varepsilon_2)$$

$$= \|D_n(\beta^\star)\| - \mathcal{O}(d\varepsilon_2).$$

Equation (30) then rewrites for any $t \in [\tau, \tau_{2,+}]$ as

$$\frac{\mathrm{d}\langle \overline{\mathsf{w}}_i(t), \overline{\mathsf{w}}_j(t) \rangle}{\mathrm{d}t} \geq (2\|D_n(\beta^\star)\| - \mathcal{O}(d\varepsilon_2)) (1 - \langle \overline{\mathsf{w}}_i(t), \overline{\mathsf{w}}_j(t) \rangle).$$

Moreover, thanks to Lemma 3, a simple algebraic manipulation yields[8] $\langle \overline{\mathsf{w}}_i(\tau), \overline{\mathsf{w}}_j(\tau) \rangle \geq 1 - \mathcal{O}\left( \lambda^\varepsilon + \sqrt{\frac{d^2 \log n}{n}} \right)$. Grönwall inequality then yields, for the considered range of parameters,

$$\langle \overline{\mathsf{w}}_i(\tau_{2,+}), \overline{\mathsf{w}}_j(\tau_{2,+}) \rangle \geq 1 - (1 - \langle \overline{\mathsf{w}}_i(\tau), \overline{\mathsf{w}}_j(\tau) \rangle) e^{-(2\|D_n(\beta^\star)\| - \mathcal{O}(d\varepsilon_2))(\tau_{2,+} - \tau)}$$

$$\geq 1 - \mathcal{O}\left( \lambda^\varepsilon + \sqrt{\frac{d^2 \log n}{n}} \right) \frac{\lambda^{2-4\varepsilon}}{\varepsilon_2} e^{\mathcal{O}\left( d\varepsilon_2 \ln(\frac{\varepsilon_2}{\lambda^{2-4\varepsilon}}) \right)}$$

$$\geq 1 - \mathcal{O}\left( \frac{\lambda}{\varepsilon_2} \right) \lambda^{-(2-4\varepsilon)\mathcal{O}(d\varepsilon_2)}.$$

The second inequality comes from the bound on $\tau_{2,+} - \tau$ in Equation (29). The third one comes from the fact that $\varepsilon \leq \frac{1}{3}$ and $\varepsilon_2 \ln(\varepsilon_2) = \mathcal{O}(1)$. Noticing that $2 - 4\varepsilon \geq 1$ finally yields the first item of Lemma 5 for a small enough $\varepsilon_2^\star = \Theta(d^{-\frac{3}{2}})$. $\qquad\square$

### E.5 PHASE 3: NEURONS FAST GROWTH

The third phase is defined for some $\varepsilon_3 > 0$ and $\delta_3$ by the following stopping time, for any $\circ \in \{+, -\}$:

$$\tau_{3,\circ} = \inf\{t \geq \tau_{2,\circ} \mid \|\hat{\beta}_\circ(t) - \beta_{n,\circ}\|_{\Sigma_{n,\circ}} \leq \varepsilon_3 \text{ or } \exists i \in \mathcal{I}_\circ, k \in \mathcal{S}_\circ, \langle \mathsf{w}_i(t), \frac{x_k}{\|x_k\|} \rangle \leq \delta_3\},$$

$$\text{where} \quad \hat{\beta}_\circ(t) = \sum_{i \in \mathcal{I}_\circ} \mathsf{a}_i(t) \mathsf{w}_i(t).$$

**Lemma 6.** *If Assumption 1 holds, for any $\varepsilon \in (0, \frac{1}{4})$, there exist $\lambda^\star = \Theta(\frac{1}{d})$, $\varepsilon_2^\star = \Theta(d^{-\frac{3}{2}})$, $n^\star = \Theta(d^3 \log d)$, $\alpha_0 = \Theta(1)$, $\delta_3 = \Theta(\frac{1}{\sqrt{d}})$ and $\varepsilon_3^\star = \Theta(1)$ such that for any $\lambda \leq \lambda^\star$, $n \geq n^\star$, $\circ \in \{+, -\}$, $\varepsilon_2 \in [\lambda^{2-4\varepsilon}, \varepsilon_2^\star]$ and $\varepsilon_3 \in [\lambda^{\alpha_0 \varepsilon \varepsilon_2}, \varepsilon_3^\star]$, with probability $1 - \mathcal{O}\left( \frac{d^2}{n} + \frac{1}{2^m} \right)$, $\tau_{3,\circ} < +\infty$ and*

    *1. neurons in $\mathcal{I}_\circ$ are in the same cone as $\circ\beta^\star$ for any $t \in [\tau, \tau_{2,\circ}]$:*

$$\forall i \in \mathcal{I}_\circ, \quad \min_{k \in \mathcal{S}_\circ} \langle \overline{\mathsf{w}}_i(t), \frac{x_k}{\|x_k\|} \rangle \geq 2\delta_3 \quad \text{and} \quad \max_{k \in \mathcal{S}_{-\circ}} \langle \overline{\mathsf{w}}_i(t), \frac{x_k}{\|x_k\|} \rangle \leq -2\delta_3.$$

*In particular, $\|\hat{\beta}_\circ(\tau_{3,\circ}) - \beta_{n,\circ}\|_{\Sigma_{n,\circ}} = \varepsilon_3$ by continuity.*

*Proof.* Similarly to the proof of Lemma 5, we assume that $\circ = +$, that the random event $\mathcal{I}_+ \neq \emptyset$, Equations (17) and (18) and the first and second items states in Lemma 5 all hold. We can first show

---

[8] A similar manipulation can be found in (Boursier and Flammarion, 2024, proof of Lemma 5).

that for any $t \in [\tau_{2,+}, \tau_{3,+}]$,

$$\sum_{i \in \mathcal{I}_+} \mathsf{a}_i(t)^2 \geq \varepsilon_2.$$

Indeed, recall that the output weights $\mathsf{a}_i$ evolve for any $t \in [\tau_{2,+}, \tau_{3,+}]$ as

$$\frac{\mathrm{d}\mathsf{a}_i(t)}{\mathrm{d}t} = \langle \mathsf{w}_i(t), D_+(t) \rangle$$

$$= \langle \mathsf{w}_i(t), D_n(\beta^\star) \rangle - \frac{1}{n} \sum_{k \in \mathcal{S}_+} h_{\vartheta(t)}(x_k) \langle \mathsf{w}_i(t), x_k \rangle$$

$$\geq \mathsf{a}_i(t) \left( \frac{1}{n} \sum_{k \in \mathcal{S}_+} \langle \overline{\mathsf{w}}_i(t), x_k \rangle \langle \beta^\star, x_k \rangle + \frac{1}{n} \sum_{k \in \mathcal{S}_+} \langle \overline{\mathsf{w}}_i(t), \eta_k x_k \rangle - \mathcal{O}\left( d \sum_{i \in \mathcal{I}_+} \mathsf{a}_i(t)^2 \right) \right). \tag{31}$$

The last inequality comes from the fact that $\frac{\sum_{k \in \mathcal{S}_+} \|x_k\|^2}{n} = \mathcal{O}(d)$. From then, note that $\frac{1}{n} \sum_{k \in \mathcal{S}_+} \langle \overline{\mathsf{w}}_i(t), x_k \rangle \langle \beta^\star, x_k \rangle \geq \Omega(\delta_3 \sqrt{d})$ during this phase. Moreover, using Chebyshev inequality, we can show for any $z > 0$ that with probability at least $1 - \mathcal{O}(\frac{d}{z^2 n})$

$$\frac{1}{n} \left\| \sum_{k \in \mathcal{S}_+} \eta_k x_k \right\|_2 \leq z. \tag{32}$$

Taking a small enough $z = \Theta(\delta_3)$, Equation (32) holds with probability $1 - \mathcal{O}\left( \frac{d}{\delta_3^2 n} \right)$ and, along Equation (31), this implies that for any $t \in [\tau_{2,+}, \tau_{3,+}]$ and $i \in \mathcal{I}_+$:

$$\frac{\mathrm{d}\mathsf{a}_i(t)}{\mathrm{d}t} \geq \mathsf{a}_i(t) \left( \Omega(\delta_3) - \mathcal{O}\left( d \sum_{i \in \mathcal{I}_+} \mathsf{a}_i(t)^2 \right) \right).$$

In particular, there exists $r = \Theta(\frac{\delta_3}{d})$ such that if $\sum_{i \in \mathcal{I}_+} \mathsf{a}_i(t)^2 \leq r$, all the $\mathsf{a}_i(t)$ are increasing. Moreover thanks to Lemma 5, $\sum_{i \in \mathcal{I}_+} \mathsf{a}_i(\tau_{2,+})^2 = \varepsilon_2$. As $\delta_3 = \Theta(\frac{1}{\sqrt{d}})$, we can choose $\varepsilon_2^\star = \Theta(d^{-\frac{3}{2}})$ small enough so that during the third phase,

$$\sum_{i \in \mathcal{I}_+} \mathsf{a}_i(t)^2 \geq \varepsilon_2. \tag{33}$$

Now note that by definition of $\beta_{n,+}$,

$$D_+(t) = -\frac{1}{n} \sum_{k \in \mathcal{S}_+} x_k x_k^\top \hat{\beta}_+(t) - x_k y_k$$

$$= -\Sigma_{n,+}(\hat{\beta}_+(t) - \beta_{n,+}) \tag{34}$$

As a consequence, $\hat{\beta}_+(t)$ evolves as follows:

$$\frac{\mathrm{d}\hat{\beta}_+(t)}{\mathrm{d}t} = \sum_{i \in \mathcal{I}_+} \left( \mathsf{a}_i(t)^2 \mathbf{I}_d + \mathsf{w}_i(t)\mathsf{w}_i(t)^\top \right) D_+(t)$$

$$= -\left( \sum_{i \in \mathcal{I}_+} \mathsf{a}_i(t)^2 \mathbf{I}_d + \sum_{i \in \mathcal{I}_+} \mathsf{w}_i(t)\mathsf{w}_i(t)^\top \right) \Sigma_{n,+}(\hat{\beta}_+(t) - \beta_{n,+})$$

In particular, this implies:

$$\frac{1}{2}\frac{\mathrm{d}\|\hat{\beta}_+(t) - \beta_{n,+}\|^2_{\Sigma_{n,+}}}{\mathrm{d}t} = \left\langle \frac{\mathrm{d}\hat{\beta}_+(t)}{\mathrm{d}t}, \Sigma_{n,+}(\hat{\beta}_+(t) - \beta_{n,+}) \right\rangle \tag{35}$$

$$= -(\hat{\beta}_+(t) - \beta_{n,+})^\top \Sigma_{n,+} \left( \sum_{i \in \mathcal{I}_+} \mathsf{a}_i(t)^2 \mathbf{I}_d + \sum_{i \in \mathcal{I}_+} \mathsf{w}_i(t)\mathsf{w}_i(t)^\top \right) \Sigma_{n,+}(\hat{\beta}_+(t) - \beta_{n,+}). \tag{36}$$

The matrix $\Sigma_{n,+}^{1/2}\left( \sum_{\mathcal{I}_+} \mathsf{a}_i(t)^2 \mathbf{I}_d + \sum_{\mathcal{I}_+} \mathsf{w}_i(t)\mathsf{w}_i(t)^\top \right)\Sigma_{n,+}^{1/2}$ is symmetric, positive definite. Thanks to Equation (33), its smallest eigenvalue is larger than $\varepsilon_2 \lambda_{\min}(\Sigma_{n,+})$, where $\lambda_{\min}(\cdot)$ denotes the smallest eigenvalue of a matrix. Using typical concentration inequalities on the empirical covariance (see e.g. Vershynin, 2018, Section 4.7), with probability $1 - \mathcal{O}\left(\frac{1}{n}\right)$, $\|\Sigma_{n,+} - \frac{\Sigma}{2}\|_{\mathrm{op}} = \mathcal{O}\left( \sqrt{\frac{d + \log n}{n}} \right)$. With the fourth point in Assumption 1, we then have for a large enough $n^\star = \Theta(d^3 \log d)$ and with probability $1 - \mathcal{O}\left(\frac{1}{n}\right)$,

$$\mathcal{O}(1) \geq \lambda_{\max}(\Sigma_{n,+}) \geq \lambda_{\min}(\Sigma_{n,+}) \geq \Omega(1)$$

$$\text{and} \quad \frac{\lambda_{\max}(\Sigma_{n,+})}{\lambda_{\min}(\Sigma_{n,+})} = \frac{\lambda_{\max}(\Sigma)}{\lambda_{\min}(\Sigma)} + \mathcal{O}\left( \sqrt{\frac{d + \log n}{n}} \right), \tag{37}$$

where $\lambda_{\max}(\cdot)$ denotes the largest eigenvalue.

Assume Equation (37) holds in the following, so that the smallest eigenvalue of $\Sigma_{n,+}^{1/2}\left( \sum_{\mathcal{I}_+} \mathsf{a}_i(t)^2 \mathbf{I}_d + \sum_{\mathcal{I}_+} \mathsf{w}_i(t)\mathsf{w}_i(t)^\top \right)\Sigma_{n,+}^{1/2}$ is larger than a term of order $\varepsilon_2$. As a consequence, Equation (36) yields

$$\frac{1}{2}\frac{\mathrm{d}\|\hat{\beta}_+(t) - \beta_{n,+}\|^2_{\Sigma_{n,+}}}{\mathrm{d}t} \leq -\Omega(\varepsilon_2)\|\hat{\beta}_+(t) - \beta_{n,+}\|^2_{\Sigma_{n,+}}.$$

Since the third phase ends if $\|\hat{\beta}_+(t) - \beta_{n,+}\|^2_{\Sigma_{n,+}}$ becomes smaller than $\varepsilon_3^2$, this yields:

$$\tau_{3,+} - \tau_{2,+} = \mathcal{O}\left( \frac{1}{\varepsilon_2} \ln(\frac{1}{\varepsilon_3}) \right). \tag{38}$$

Now recall that for any $i, j \in \mathcal{I}_+$,

$$\frac{\mathrm{d}(1 - \langle \overline{\mathsf{w}}_i(t), \overline{\mathsf{w}}_j(t) \rangle)}{\mathrm{d}t} = -\langle D_+(t), \overline{\mathsf{w}}_i(t) + \overline{\mathsf{w}}_j(t) \rangle (1 - \langle \overline{\mathsf{w}}_i(t), \overline{\mathsf{w}}_j(t) \rangle)$$

$$\leq 2\|D_+(t)\|_2 (1 - \langle \overline{\mathsf{w}}_i(t), \overline{\mathsf{w}}_j(t) \rangle).$$

Notice from Equation (34) and the previous discussion that $\|D_+(t)\|_2 = \mathcal{O}(1)$. As a consequence, a simple Grönwall inequality with Equation (38) yields that for any $t \in [\tau_{2,+}, \tau_{3,+}]$:

$$\langle \overline{\mathsf{w}}_i(t), \overline{\mathsf{w}}_j(t) \rangle \geq 1 - (1 - \langle \overline{\mathsf{w}}_i(\tau_{2,+}), \overline{\mathsf{w}}_j(\tau_{2,+}) \rangle) \exp((t - \tau_{2,+})\mathcal{O}(1))$$

$$\geq 1 - \mathcal{O}\left( \frac{\lambda^{\frac{1}{2}}}{\varepsilon_2} \right) \exp\left( \mathcal{O}\left( \frac{1}{\varepsilon_2} \ln(\frac{1}{\varepsilon_3}) \right) \right)$$

$$\geq 1 - \mathcal{O}\left( \lambda^{\frac{1}{2} - \varepsilon} \right).$$

The second inequality comes from the value of $(1 - \langle \overline{\mathsf{w}}_i(\tau_{2,+}), \overline{\mathsf{w}}_j(\tau_{2,+}) \rangle)$, thanks to Lemma 5. The last one comes from our choice of $\varepsilon_3$ for a large enough $\alpha_0 = \Theta(1)$.

In particular, this last inequality can be used to show[9] that for any $i, j \in \mathcal{I}_+$ and $t \in [\tau_{2,+}, \tau_{3,+}]$, $\overline{\mathsf{w}}_i(t) = \overline{\mathsf{w}}_j(t) + \mathcal{O}\left(\lambda^{\frac{1-2\varepsilon}{4}}\right)$. In particular, this yields for any $i \in \mathcal{I}_+$ and $t \in [\tau_{2,+}, \tau_{3,+}]$

$$\hat{\beta}_+(t) = \sum_{j \in \mathcal{I}_+} \mathsf{a}_j(t)^2 \overline{\mathsf{w}}_j(t) \tag{39}$$

$$= \left(\sum_{j \in \mathcal{I}_+} \mathsf{a}_j(t)^2\right) \left(\overline{\mathsf{w}}_i(t) + \mathcal{O}\left(\lambda^{\frac{1-2\varepsilon}{4}}\right)\right). \tag{40}$$

Since $\|\overline{\mathsf{w}}_i(t)\|_2 = 1 - \mathcal{O}\left(\lambda^{\frac{1}{2}-\varepsilon}\right)$, this last equality actually yields the following comparison for $t \in [\tau_{2,+}, \tau_{3,+}]$:

$$\|\hat{\beta}_+(t)\|_2 \leq \sum_{j \in \mathcal{I}_+} \mathsf{a}_j(t)^2 \leq (1 + \mathcal{O}\left(\lambda^{\frac{1-2\varepsilon}{4}}\right))\|\hat{\beta}_+(t)\|_2. \tag{41}$$

In particular, since $\|\hat{\beta}_+(t)\|_2 = \mathcal{O}(1)$, this yields $\sum_{j \in \mathcal{I}_+} \mathsf{a}_j(t)^2 = \mathcal{O}(1)$.

From there, for any $x_k \in \mathcal{S}_+$ and $i \in \mathcal{I}_+$, $\langle \overline{\mathsf{w}}_i(t), x_k \rangle$ evolves as follows during the third phase

$$\frac{\mathrm{d}\langle \overline{\mathsf{w}}_i(t), \frac{x_k}{\|x_k\|}\rangle}{\mathrm{d}t} = \langle D_+(t), \frac{x_k}{\|x_k\|}\rangle - \langle D_+(t), \overline{\mathsf{w}}_i(t)\rangle\langle \overline{\mathsf{w}}_i(t), \frac{x_k}{\|x_k\|}\rangle$$

$$= \langle \beta_{n,+} - \hat{\beta}_+(t), \Sigma_{n,+} \frac{x_k}{\|x_k\|}\rangle - \mathcal{O}\left(\langle \overline{\mathsf{w}}_i(t), \frac{x_k}{\|x_k\|}\rangle\right)$$

$$= \frac{1}{2}\langle \beta_{n,+} - \hat{\beta}_+(t), \frac{x_k}{\|x_k\|}\rangle + \langle (\Sigma_{n,+} - \frac{\mathbf{I}_d}{2})(\beta_{n,+} - \hat{\beta}_+(t)), \frac{x_k}{\|x_k\|}\rangle - \mathcal{O}\left(\langle \overline{\mathsf{w}}_i(t), \frac{x_k}{\|x_k\|}\rangle\right).$$

Note that

$$\beta_{n,+} = \beta^\star + \frac{\Sigma_{n,+}^{-1}}{n} \sum_{k \in \mathcal{S}_+} \eta_k x_k.$$

This then yields, thanks to Equation (37)

$$\frac{\mathrm{d}\langle \overline{\mathsf{w}}_i(t), \frac{x_k}{\|x_k\|}\rangle}{\mathrm{d}t} \geq \frac{1}{2}\langle \beta^\star - \hat{\beta}_+(t), \frac{x_k}{\|x_k\|}\rangle + \langle (\Sigma_{n,+} - \frac{\mathbf{I}_d}{2})(\beta_{n,+} - \hat{\beta}_+(t)), \frac{x_k}{\|x_k\|}\rangle$$

$$- \mathcal{O}\left(\frac{1}{n}\|\sum_{k \in \mathcal{S}_+} \eta_k x_k\|_2\right) - \mathcal{O}\left(\langle \overline{\mathsf{w}}_i(t), \frac{x_k}{\|x_k\|}\rangle\right). \tag{42}$$

From there, thanks to the third point of Assumption 1 and Equation (40):

$$\langle \beta^\star - \hat{\beta}_+(t), \frac{x_k}{\|x_k\|}\rangle \geq \frac{c}{\sqrt{d}} - \mathcal{O}\left(\langle \overline{\mathsf{w}}_i(t), \frac{x_k}{\|x_k\|}\rangle\right) - \mathcal{O}\left(\lambda^{\frac{1-2\varepsilon}{4}}\right). \tag{43}$$

---

[9]For that, we decompose $\overline{\mathsf{w}}_i = \alpha_{ij}\overline{\mathsf{w}}_j + u_{ij}$ with $u_{ij} \perp \overline{\mathsf{w}}_j$ and show that $\alpha_{ij} = 1 - \mathcal{O}\left(\lambda^{\frac{1}{2}-\varepsilon}\right)$ and $\|u_{ij}\|^2 = \mathcal{O}\left(\lambda^{\frac{1}{2}-\varepsilon}\right)$.

Additionally, using the fact that $\|\beta_{n,+} - \hat{\beta}_+(t)\|_{\Sigma_{n,+}}$ is decreasing over time and smaller than $\|\beta_{n,+}\|_{\Sigma_{n,+}} + \mathcal{O}(\lambda^{2-4\varepsilon})$ at the beginning of the second phase,

$$\langle (\Sigma_{n,+} - \frac{\mathbf{I}_d}{2})(\beta_{n,+} - \hat{\beta}_+(t)), \frac{x_k}{\|x_k\|} \rangle \geq -\left\| \Sigma_{n,+} - \frac{\mathbf{I}_d}{2} \right\|_{\mathrm{op}} \|\beta_{n,+} - \hat{\beta}_+(t)\|_2$$

$$\geq -\frac{1}{2} \|\Sigma - \mathbf{I}_d\|_{\mathrm{op}} \sqrt{\frac{1}{\lambda_{\min}(\Sigma_{n,+})}} \|\beta_{n,+} - \hat{\beta}_+(t)\|_{\Sigma_{n,+}} - \mathcal{O}\left( \sqrt{\frac{d + \log n}{n}} \right)$$

$$\geq -\frac{1}{2} \|\Sigma - \mathbf{I}_d\|_{\mathrm{op}} \sqrt{\frac{1}{\lambda_{\min}(\Sigma_{n,+})}} \left( \|\beta_{n,+}\|_{\Sigma_{n,+}} + \mathcal{O}(\lambda^{2-4\varepsilon}) \right) - \mathcal{O}\left( \sqrt{\frac{d + \log n}{n}} \right)$$

$$\geq -\frac{1}{2} \|\Sigma - \mathbf{I}_d\|_{\mathrm{op}} \sqrt{\frac{\lambda_{\max}(\Sigma_{n,+})}{\lambda_{\min}(\Sigma_{n,+})}} \|\beta_{n,+}\|_2 - \mathcal{O}\left( \lambda^{2-4\varepsilon} + \sqrt{\frac{d + \log n}{n}} \right)$$

$$\geq -\frac{1}{2} \sqrt{\frac{\lambda_{\max}(\Sigma_{n,+})}{\lambda_{\min}(\Sigma_{n,+})}} \|\Sigma - \mathbf{I}_d\|_{\mathrm{op}} \|\beta^\star\|_2$$

$$- \mathcal{O}\left( \lambda^{2-4\varepsilon} + \frac{1}{n}\| \sum_{k \in \mathcal{S}_+} \eta_k x_k \|_2 + \sqrt{\frac{d + \log n}{n}} \right)$$

Now using Equation (37) and the fourth point of Assumption 1, note that

$$\sqrt{\frac{\lambda_{\max}(\Sigma_{n,+})}{\lambda_{\min}(\Sigma_{n,+})}} \leq 2 + \mathcal{O}\left( \sqrt{\frac{d + \log n}{n}} \right).$$

So that the previous inequality yields

$$\langle (\Sigma_{n,+} - \frac{\mathbf{I}_d}{2})(\beta_{n,+} - \hat{\beta}_+(t)), \frac{x_k}{\|x_k\|} \rangle \geq -\|\Sigma - \mathbf{I}_d\|_{\mathrm{op}} \|\beta^\star\|_2 - \mathcal{O}\left( \lambda^{2-4\varepsilon} + \frac{1}{n}\| \sum_{k \in \mathcal{S}_+} \eta_k x_k \|_2 + \sqrt{\frac{d + \log n}{n}} \right).$$
(44)

Finally, thanks to Equation (32), $\frac{1}{n}\| \sum_{k \in \mathcal{S}_+} \eta_k x_k \|_2 \leq z'$ with probability at least $1 - \mathcal{O}(\frac{d}{z'^2 n})$. Using Equations (43) and (44) in Equation (42) finally yields for the third phase:

$$\frac{\mathrm{d}\langle \overline{\mathrm{w}}_i(t), \frac{x_k}{\|x_k\|} \rangle}{\mathrm{d}t} \geq \frac{c}{2\sqrt{d}} - \|\Sigma - \mathbf{I}_d\|_{\mathrm{op}}\|\beta^\star\|_2 - \mathcal{O}\left( \lambda^{\frac{1-2\varepsilon}{4}} + z' + \sqrt{\frac{d + \log n}{n}} \right) - \mathcal{O}\left( \langle \overline{\mathrm{w}}_i(t), \frac{x_k}{\|x_k\|} \rangle \right).$$

Thanks to the fourth point of Assumption 1, $\frac{c}{2\sqrt{d}} - \|\Sigma - \mathbf{I}_d\|_{\mathrm{op}}\|\beta^\star\|_2 > 0$, so that we can choose $\lambda^\star, z' = \Theta(1)$ small enough and $n^\star = \Theta(d^3 \log d)$ large enough so that the previous inequality becomes, with probability at least $1 - \mathcal{O}(\frac{d}{n})$

$$\frac{\mathrm{d}\langle \overline{\mathrm{w}}_i(t), \frac{x_k}{\|x_k\|} \rangle}{\mathrm{d}t} \geq \Omega(\frac{1}{\sqrt{d}}) - \mathcal{O}\left( \langle \overline{\mathrm{w}}_i(t), \frac{x_k}{\|x_k\|} \rangle \right).$$

A simple Grönwall argument with the second point of Lemma 5 then implies that for any $t \in [\tau_{2,+}, \tau_{3,+}]$, $i \in \mathcal{I}_+$ and $k \in \mathcal{S}_+$,

$$\langle \overline{\mathrm{w}}_i(t), \frac{x_k}{\|x_k\|} \rangle \geq \Omega(\frac{1}{\sqrt{d}}).$$

Since the term $\Omega(\frac{1}{\sqrt{d}})$ here does not depend on $\delta_3$, we can choose $\delta_3 = \Theta(\frac{1}{\sqrt{d}})$ small enough so that

$$\langle \overline{\mathrm{w}}_i(t), \frac{x_k}{\|x_k\|} \rangle \geq 2\delta_3.$$

We can show similarly for $k \in \mathcal{S}_-$, so that point 1 in Lemma 6 holds, which concludes the proof. $\square$

### E.6 PHASE 4: FINAL CONVERGENCE

The last phase is defined for some $\varepsilon_4 > \varepsilon_3$ by the following stopping time, for any $\circ \in \{+, -\}$:

$$\tau_{4,\circ} = \inf\{t \geq \tau_{3,\circ} \mid \|\hat{\beta}_\circ(t) - \hat{\beta}_\circ(\tau_{3,\circ})\|_{\Sigma_{n,\circ}} \geq \varepsilon_4\}.$$

**Lemma 7.** *If Assumption 1 holds, for any $\varepsilon \in (0, \frac{1}{4})$, there exist $\lambda^\star = \Theta(\frac{1}{d})$, $\varepsilon_2^\star = \Theta(d^{-\frac{3}{2}})$, $n^\star = \Theta(d^3 \log d)$, $\alpha_0 = \Theta(1)$, $\delta_3 = \Theta(\frac{1}{\sqrt{d}})$, $\varepsilon_3^\star = \Theta(\frac{1}{\sqrt{d}})$ and $\varepsilon_4 = \Theta(\varepsilon_3^\star)$ such that for any $\lambda \leq \lambda^\star$, $n \geq n^\star$, $\circ \in \{+, -\}$, $\varepsilon_2 \in [\lambda^{2-4\varepsilon}, \varepsilon_2^\star]$ and $\varepsilon_3 \in [\lambda^{\alpha_0 \varepsilon \varepsilon_2}, \varepsilon_3^\star]$, with probability $1 - \mathcal{O}\left(\frac{d^2}{n} + \frac{1}{2^m}\right)$, $\tau_{4,\circ} = +\infty$ and*

*1. neurons in $\mathcal{I}_\circ$ are in the same cone as $\circ \beta^\star$ for any $t \geq \tau_{3,\circ}$:*

$$\forall i \in \mathcal{I}_\circ, \quad \min_{k \in \mathcal{S}_\circ} \langle \overline{w}_i(t), \frac{x_k}{\|x_k\|} \rangle > 0 \quad and \quad \max_{k \in \mathcal{S}_{-\circ}} \langle \overline{w}_i(t), \frac{x_k}{\|x_k\|} \rangle < 0.$$

*2. $\lim_{t\to\infty} \vartheta(t)$ exists and $\lim_{t\to\infty} \hat{\beta}_\circ(t) = \beta_{n,\circ}$.*

*Proof.* Similarly to the previous phases, we assume that $\circ = +$, that the random event $\mathcal{I}_+ \neq \emptyset$, Equations (17), (18) and (37) and the statements of Lemma 6 all hold.

Define in the following the positive loss $L_+$ for any $\vartheta_+ \in \mathbb{R}^{(d+1) \times \mathcal{I}_+}$ by

$$L_+(\vartheta) = \frac{1}{2n} \sum_{k \in \mathcal{S}_+} \left( \sum_{i \in \mathcal{I}+} a_i \langle w_i, x_k \rangle - y_k \right)^2.$$

Note that the autonomous system given by Equation (23) actually defines a gradient flow over $L_+$, i.e., for $\vartheta_+ = (a_i, w_i)_{i \in \mathcal{I}_+}$,

$$\frac{d\vartheta_+(t)}{dt} = -\nabla L_+(\vartheta_+(t)).$$

The main argument for this phase is to prove a local Polyak-Łojasiewicz inequality:

$$\|\nabla L_+(\vartheta_+)\|_2^2 \geq \Omega(1)(L_+(\vartheta_+) - L_{n,+}) \tag{45}$$

$$\text{for any } \vartheta_+ \text{ such that } \quad \| \sum_{i \in \mathcal{I}_+} a_i w_i - \hat{\beta}_+(\tau_{3,\circ}) \|_{\Sigma_{n,+}} \leq \varepsilon_4,$$

$$\text{where} \quad L_{n,+} = \frac{1}{2n} \sum_{k \in \mathcal{S}_+} \left( \langle \beta_{n,+}, x_k \rangle - y_k \right)^2.$$

Indeed, we can lower bound $\|\nabla L_+(\vartheta_+)\|_2$ for any such $\vartheta_+$ as follows

$$\|\nabla L_+(\vartheta_+)\|_2^2 \geq \sum_{i \in \mathcal{I}_+} \left\| \frac{\partial L_+(\vartheta_+)}{\partial w_i} \right\|^2$$

$$= \left( \sum_{i \in \mathcal{I}_+} a_i(t)^2 \right) \|D_+(t)\|_2^2$$

$$\geq \lambda_{\min}(\Sigma_{n,+}) \left( \sum_{i \in \mathcal{I}_+} a_i(t)^2 \right) \|\hat{\beta}_+ - \beta_{n,+}\|_{\Sigma_{n,+}}^2$$

where $\hat{\beta}_+ = \sum_{i \in \mathcal{I}_+} a_i w_i$. The last inequality comes from Equation (34). Note that for a small enough choice of $\varepsilon_3^\star = \mathcal{O}(1)$ and $\varepsilon_4 = \Theta(\varepsilon_3^\star)$, $\sum_{i \in \mathcal{I}_+} a_i(t)^2 = \Omega(1)$ in the considered set. Moreover, Equation (37) implies $\lambda_{\min}(\Sigma_{n,+}) = \Omega(1)$, so that

$$\|\nabla L_+(\vartheta_+)\|_2^2 \geq \Omega(1) \|\hat{\beta}_+ - \beta_{n,+}\|_{\Sigma_{n,+}}^2. \tag{46}$$

On the other hand, a simple algebraic manipulation yields for any $\vartheta_+$:

$$L_+(\vartheta_+) - L_{n,+} = \frac{1}{2n} \sum_{k \in \mathcal{S}_+} \left( \langle \hat{\beta}_+, x_k \rangle - y_k \right)^2 - (\langle \beta_{n,+}, x_k \rangle - y_k)^2$$

$$= \frac{1}{2n} \sum_{k \in \mathcal{S}_+} \left( \hat{\beta}_+ - \beta_{n,+} \right)^\top x_k - \left( x_k^\top (\hat{\beta}_+ - \beta_{n,+} + 2\beta_{n,+}) - 2y_k \right)$$

$$= \frac{1}{2} \left( \hat{\beta}_+ - \beta_{n,+} \right)^\top \Sigma_{n,+} \left( \hat{\beta}_+ - \beta_{n,+} \right) + \frac{1}{n} \mathbf{X}_{n,+} (\mathbf{X}_{n,+}^\top \beta_{n,+} - \mathbf{y}),$$

where $\mathbf{X}_{n,+}$ is the $|\mathcal{S}_+| \times d$ matrix, whose rows are given by $x_k$ for $k \in \mathcal{S}_+$. By definition of the OLS estimator $\beta_{n,+}$, $\mathbf{X}_{n,+}^\top \beta_{n,+} - \mathbf{y} = \mathbf{0}$, so that

$$L_+(\vartheta_+) - L_{n,+} = \frac{1}{2} \|\hat{\beta}_+ - \beta_{n,+}\|_{\Sigma_{n,+}}^2. \tag{47}$$

Combining Equation (46) with Equation (47) finally yields the Polyak-Łojasiewicz inequality given by Equation (45).

From there, this implies by chain rule for any $t \in [\tau_{3,+}, \tau_{4,+}]$

$$\frac{dL_+(\vartheta_+(t))}{dt} = -\|\nabla L_+(\vartheta_+)\|_2^2$$

$$\leq -\Omega(1)(L_+(\vartheta_+(t)) - L_{n,+}).$$

By Grönwall inequality, this implies for some $\nu = \Theta(1)$, for any $t \in [\tau_{3,+}, \tau_{4,+}]$

$$L_+(\vartheta_+(t)) - L_{n,+} \leq (L_+(\vartheta_+(\tau_{3,+})) - L_{n,+}) e^{-\nu(t-\tau_{3,+})}$$

$$\leq \frac{\varepsilon_3^2}{2} e^{-\nu(t-\tau_{3,+})}. \tag{48}$$

The last inequality comes from the fact that at the end of the third phase, $\|\hat{\beta}_+(t) - \beta_{n,+}\|_{\Sigma_{n,+}} = \varepsilon_3$.

We bounded by below the norm of $\nabla L_+(\vartheta_+(s))$, but it can also easily be bounded by above as

$$\|\nabla L_+(\vartheta_+(s))\|_2^2 \leq \left( \sum_{i \in \mathcal{I}_+} \mathsf{a}_i(t)^2 + \|\mathsf{w}_i(t)\|_2^2 \right) \|D_+(t)\|_2^2$$

$$\leq 2\lambda_{\max}(\Sigma_{n,+}) \left( \sum_{i \in \mathcal{I}_+} \mathsf{a}_i(t)^2 \right) \|\hat{\beta}_+(t) - \beta_{n,+}\|_{\Sigma_{n,+}}^2$$

$$\leq \mathcal{O}(1) \left( L_+(\vartheta_+(t)) - L_{n,+} \right)$$

From there, the variation of $\vartheta_+(t)$ can easily be bounded for any $t \in [\tau_{3,+}, \tau_{4,+}]$ as

$$\|\vartheta_+(t) - \vartheta(\tau_{3,+})\|_2 \leq \int_{\tau_{3,+}}^t \|\nabla L_+(\vartheta_+(s))\| ds$$

$$\leq \mathcal{O}(1) \varepsilon_3 \int_0^{t-\tau_{3,+}} e^{-\frac{\nu}{2} s} ds$$

$$\leq \mathcal{O}(\varepsilon_3). \tag{49}$$

Moreover, note that

$$\hat{\beta}_+(t) - \hat{\beta}_\circ(\tau_{3,+}) = \sum_{\mathcal{I}_+} (\mathsf{a}_i(t) - \mathsf{a}_i(\tau_{3,+})) \mathsf{w}_i(\tau_{3,+}) + \sum_{\mathcal{I}_+} (\mathsf{w}_i(t) - \mathsf{w}_i(\tau_{3,+})) \mathsf{a}_i(\tau_{3,+}).$$

In particular,

$$\|\hat{\beta}_+(t) - \hat{\beta}_\circ(\tau_{3,+})\|_2 \leq \sum_{\mathcal{I}_+} |\mathsf{a}_i(t) - \mathsf{a}_i(\tau_{3,+})| \|\mathsf{w}_i(\tau_{3,+})\|_2 + \sum_{\mathcal{I}_+} \|\mathsf{w}_i(t) - \mathsf{w}_i(\tau_{3,+})\|_2 \mathsf{a}_i(\tau_{3,+})$$

$$\leq \sqrt{\sum_{\mathcal{I}_+} (\mathsf{a}_i(t) - \mathsf{a}_i(\tau_{3,+}))^2} \sqrt{\sum_{\mathcal{I}_+} \|\mathsf{w}_i(\tau_{3,+})\|_2^2} + \sqrt{\sum_{\mathcal{I}_+} \|\mathsf{w}_i(t) - \mathsf{w}_i(\tau_{3,+})\|_2^2} \sqrt{\sum_{\mathcal{I}_+} \mathsf{a}_i(\tau_{3,+})^2}$$

$$\leq \mathcal{O}(1) \|\vartheta(t) - \vartheta(\tau_{3,+})\|_2$$

$$\leq \mathcal{O}(\varepsilon_3).$$

We can thus choose $\varepsilon_3^\star = \mathcal{O}(1)$ and $\varepsilon_4 = \Theta(\varepsilon_3^\star)$ small enough such that Equation (46) still holds, but $\varepsilon_4$ large enough with respect to $\varepsilon_3^\star$ such that the previous inequality ensures for any $t \in [\tau_{3,+}, \tau_{4,+}]$:

$$\|\hat{\beta}_+(t) - \hat{\beta}_\circ(\tau_{3,+})\|_{\Sigma_{n,+}} \leq \frac{\varepsilon_4}{2}.$$

In particular, this implies that $\tau_{4,+} = +\infty$. Since $\vartheta_+(t)$ has finite variation (Equation 49), this also implies that $\lim_{t\to\infty} \vartheta_+(t)$ exists. The same holds for $\vartheta_-(t)$ by symmetric arguments, so that $\lim_{t\to\infty} \vartheta(t)$ exists. Moreover, Equations (47) and (48) imply that

$$\lim_{t\to\infty} \hat{\beta}_+(t) = \beta_{n,+}.$$

This yields the second point of Lemma 7.

It now remains to prove the first point of Lemma 7. Note that for any $t \geq \tau_{3,+}$ and $i \in \mathcal{I}_+$:

$$\|\bar{\mathsf{w}}_i(t) - \bar{\mathsf{w}}_i(\tau_{3,+})\|_2 \leq 2 \int_{\tau_{3,+}}^t \|D_+(s)\|_2 \mathrm{d}s$$

$$\leq \mathcal{O}(\varepsilon_3).$$

Thanks to the first point of Lemma 6, we can choose $\varepsilon_3^\star = \Theta(\frac{1}{\sqrt{d}})$ small enough so that for any $t \geq \tau_{3,+}$ and $i \in \mathcal{I}_+$:

$$\min_{k \in \mathcal{S}_+} \langle \bar{\mathsf{w}}_i(t), \frac{x_k}{\|x_k\|} \rangle > 0 \quad \text{and} \quad \max_{k \in \mathcal{S}_-} \langle \bar{\mathsf{w}}_i(t), \frac{x_k}{\|x_k\|} \rangle < 0,$$

which concludes the proof of Lemma 7. $\qquad\square$

*Proof of Theorem 2.* We can conclude the proof of Theorem 2 by noticing that we can indeed choose $\varepsilon, \varepsilon_2, \varepsilon_3, \varepsilon_4$ such that for any $\lambda \leq \lambda^\star = \Theta(\frac{1}{d})$ and $n \geq n^\star = \Theta_\mu(d^3 \log d)$, with probability $1 - \mathcal{O}\left(\frac{d^2}{n} + \frac{1}{2^m}\right)$, the statements of Lemmas 3 and 5 to 7 all simultaneously hold. In particular, the stopping times $T_+$ and $T_-$ defined in Lemma 4 are infinite. Lemma 4 then implies that for any $t \geq \tau$, $\vartheta(t) = \theta(t)$. From then, Lemma 7 implies Theorem 2. $\qquad\square$

