# OpenReview forum: "Simplicity Bias and Optimization Threshold in Two-Layer Networks"
_ICLR.cc/2025/Conference — Submitted to ICLR 2025_

### Official Review · Reviewer_uxcs · 2024-11-01

**Soundness:** 3
**Presentation:** 2
**Contribution:** 2
**Rating:** 5
**Confidence:** 5

**Summary:**

This paper studies two-layer ReLU networks trained on a noisy linear regression task. By analyzing the training dynamics, the authors show that the two-layer ReLU network learns a nearly linear estimator. The authors use this analysis to illustrate an optimization threshold phenomenon that was observed and attracted interest in practical settings.

**Strengths:**

- The optimization threshold occurring in two-layer ReLU networks trained on a linear task is an interesting and somewhat surprising result. There is not yet much theoretical work on optimization threshold. So theoretical insight on this topic is new.
- The main result is laid out with explicit assumptions and detailed discussions of its implications and limitations.

**Weaknesses:**

- It seems to be a well-known opinion [1-3] that the early alignment, learning sparse features, or simplicity bias is a double-edged sword and sometimes leads to overfitting [1] instead of good generalization. This paper says the simplicity bias is beneficial, at least in this case. So I wonder how this paper fits into previous results with the different claim. I am especially interested in the authors' comparison with [1,2] because they are also theoretical works on two-layer ReLU networks.
- The dataset this paper studies seems quite similar to [4], which also assumed symmetric input distribution and a linear target. So I am wondering if this paper addresses a more challenging setup and in what ways the technical contribution goes beyond [4].
- The flow of the paper may benefit from a clearer structure. I felt I had to jump back and forth because the main theorem is immediately followed by discussions and then its proof sketch. Also, there is no related work section, though there are scattered discussions of related work throughout the paper.
- The authors motivate their problem by introducing the optimization threshold phenomenon in transformers and diffusion models. Although these models contain fully-connected ReLU layers, the mechanism in a two-layer ReLU network doesn't necessarily explain phenomena in transformers and diffusion models. For instance, one plausible possibility is that the attention layers in transformers play a key role in inducing this phenomenon rather than the ReLU layers. So I am not convinced that the ReLU network problem is well motivated by a phenomenon observed in transformers and diffusion models, despite the possibility of other valid motivations that weren't stated.

[1] Petrini et al. "Learning sparse features can lead to overfitting in neural networks." NeurIPS 2022

[2] Frei et al. "The double-edged sword of implicit bias: Generalization vs. robustness in relu networks." NeurIPS 2023.

[3] Shah et al. "The pitfalls of simplicity bias in neural networks." NeurIPS 2020.

[4] Lyu et al. "Gradient descent on two-layer nets: Margin maximization and simplicity bias."  NeurIPS 2021.

**Questions:**

- In line 124, does this paper consider stochastic gradient descent or just gradient descent/flow? I imagine if SGD instead of GD is used, the data samples in each minibatch won't always satisfy Assumption 1?
- In line 135, what distribution are $(\tilde a_j,\tilde w_j)$ drawn i.i.d. from?
- In line 153, the definition of $\mathfrak D_n(w,\theta)$ doesn't look right. There are two left curly brackets but only one right curly bracket.
- In Figure 1, it was hard to spot $\sigma^2$ in both plots. Perhaps it's more visible if $\sigma^2$ is a tick on the y axis.

I'd be happy to raise score if the questions are properly addressed.

---

> ### Author Response · Authors · 2024-11-22
> **Rebuttal to reviewer uxcs 1/2**
>
> We thank the reviewer for his insightful feedback. We now answer your different concerns.
>
> > It seems to be a well-known opinion [1-3] that the early alignment, learning sparse features, or simplicity bias is a double-edged sword and sometimes leads to overfitting [1] instead of good generalization.
>
> Implicit or simplicity bias can have both positive and negative effects, depending on the data distribution. For instance, Ohad Shamir in The Implicit Bias of Benign Overfitting  demonstrates cases where implicit bias can be detrimental to generalization, but also highlights situations where it is beneficial. Whether this bias helps generalization in practice depends heavily on the true data distribution. Our work specifically provides a theoretical explanation for the simplicity biases observed in prior empirical studies on in-context learning and diffusion, where this bias appears to be beneficial. Our conclusions align with these findings.
>
> ## Comparison with [1]
> [1] pursues a very different goal: demonstrating that the Neural Tangent Kernel interpolator can, in certain situations, generalize better than the feature learning regime (the regime considered in our work). Most importantly, in [1], the interpolating solution still generalizes in the feature learning regime, with the generalization error decreasing to 0 as $n\to\infty$, albeit at a slower rate than the NTK interpolator.
>
> In contrast, in our work, interpolators (both NTK and feature learning) have a non-zero generalization error, even as $n\to\infty$. The novelty of our setting in the theoretical literature lies in demonstrating that avoiding interpolation is necessary for good generalization, (with the exception of very unnatural interpolators, such as linear regression with infinitely small interpolating spikes). To our knowledge, similar properties in prior theoretical work have only been explored through early stopping in training, as mentioned in the discussion section.
>
> ## Comparison with [2]
> [2] describes a benign overfitting phenomenon, where parameters converge to an interpolating solution that still generalizes well. They also show that this solution is not robust to small input perturbations.
> As mentioned in the discussion section, our contribution differs from benign overfitting: in our case, the parameters do not overfit, while any natural interpolating solution fails to generalize. The differing conclusions about interpolating solutions (i.e., whether they generalize or not) stem from the difference in regimes of dimension. Benign overfitting is specific to large-dimensional settings, whereas our setting does not require such assumptions.
>
> ## Comparison with [3]
> In [3], poor generalization is not a result of overfitting, but of how the overfitting occurs (i.e., the implicit or simplicity bias). Notably, [3] identifies a natural interpolating solution that generalizes perfectly. By contrast, in our setup, only highly unnatural interpolators generalize effectively.
>
> > The dataset this paper studies seems quite similar to [4]
>
> The work [4] is fundamentally different from ours, both technically and conceptually. They show convergence to a data interpolator, which would result in poor generalization in our setup. In contrast, our key contribution is providing an example of convergence to a non-interpolating solution that generalizes well.
>
> This difference arises because, in [4], a linear estimator overfits the data (e.g., linearly separable data in classification), whereas in our case, overfitting occurs only with more complex solutions.
>
> Moreover, [4] assumes symmetric data, while we only require a symmetric distribution.  Although these are equivalent when $n=\infty$, a main technical challenge in our work stems from the fact that optimization trajectories differ significantly when $n$ is finite, particularly since the width $m$ can be much larger than $n$.
>
> > The flow of the paper may benefit from a clearer structure
>
> While there is no dedicated related work section, we extensively discuss related works and their connections to our study, particularly in the introduction and discussion sections. We chose this structure to maintain a better flow for the paper. Given the breadth and diversity of the related literature, we preferred to address it directly where it is most relevant, such as in the various paragraphs of the discussion section.
> If the reviewer feels it is necessary, we can add a related work section in the appendix to summarize these comparisons.
>
> Additionally, in the revised version, we have switched the Discussion and Sketch of Proof sections to further improve clarity.

---

> > ### Author Response · Authors · 2024-11-22
> > **Rebuttal to reviewer uxcs 2/2**
> >
> > > the mechanism in a two-layer ReLU network doesn't necessarily explain phenomena in transformers and diffusion models
> >
> > As explained in our general response, it remains unclear how our theoretical settings extend to the more complex architectures used in practice, a limitation shared broadly in Deep Learning theory. While it is often assumed that such phenomena arise from attention layers, our goal is to challenge this assumption. We propose that the phenomena observed in Raventos et al. (2024) and Kadkhodaie et al. (2023) may instead stem from non-linear fully connected layers. Notably, Kadkhodaie et al. (2023) use a diffusion model without any attention layers. However, it is also possible that attention layers play a role in this phenomenon, e.g., in leading to a smaller dependency of the sample complexity threshold $n^{\star}$ with $d$.
> > While there is a tendency to attribute all non-trivial behaviors in transformers to attention, we aim to nuance this view by demonstrating that some of these phenomena also occur in simpler, attention-free architectures. This perspective aligns with recent experimental work suggesting that RNNs [5] or MLP mixer [6] can achieve transformer-like performance.
> >
> > > In line 124, does this paper consider stochastic gradient descent or just gradient descent/flow?
> >
> > There might be a confusion, Assumption 1 is about the training data distribution. The randomness of SGD will be over the random minibatch, sampled uniformly over the $n$ data in the training set. Assumption 1 is not about the data of each minibatch.
> > Moreover, this paper considers gradient flow. Gradient Flow is a first order approximation (as the learning rate goes to 0) of both GD and SGD (see eg [7]), and is often considered in the theoretical literature as it yields a simpler analysis.
> >
> > Note that we used SGD in our experiments.
> >
> > > In line 135, what distribution are (tilde{a}_j, tilde{w}_j) drawn i.i.d. from?
> >
> > They can be drawn i.i.d. from any distribution satisfying equation (3). Typical distribution satisfying that is $\tilde{w}_j$ uniformly at random in the unit ball, and $\tilde{a}_j$ a Rademacher variable (ie +1 or -1 with proba 1/2)
> >
> >
> > -----------------------
> >
> > # References
> >
> > [1] Petrini et al. "Learning sparse features can lead to overfitting in neural networks." NeurIPS 2022
> >
> > [2] Frei et al. "The double-edged sword of implicit bias: Generalization vs. robustness in relu networks." NeurIPS 2023.
> >
> > [3] Shah et al. "The pitfalls of simplicity bias in neural networks." NeurIPS 2020.
> >
> > [4] Lyu et al. "Gradient descent on two-layer nets: Margin maximization and simplicity bias." NeurIPS 2021.
> >
> > [5] Feng, Leo, et al. "Were RNNs All We Needed?." arXiv preprint arXiv:2410.01201 (2024).
> >
> > [6] Tolstikhin, Ilya O., et al. "Mlp-mixer: An all-mlp architecture for vision." Advances in neural information processing systems 34 (2021)
> >
> > [7] Q. Li et. al. Stochastic modified equations and dynamics of stochastic gradient algorithms i: Mathematical foundations. The Journal of Machine Learning Research, 2019.

---

> > > ### Comment · Reviewer_uxcs · 2024-11-26
> > >
> > > Thank you for your detailed response. Though I acknowledge some merit in this work, I still have concerns with potential overstatement issues in the paper and the rebuttal.
> > >
> > > I agree that two-layer ReLU networks may share some similarities on the behavioral level with the transformers and diffusion models regarding optimization threshold. But they provide limited explanatory power at a mechanistic level, as two-layer ReLU networks and transformers/diffusion differ in numerous aspects, including the architecture, data, initialization, optimizer, loss, etc. I agree with the authors that such gaps are common in theoretical works and do not necessarily constitute a weakness. However, motivating the problem by focusing on the connections but not adequately acknowledging these gaps may result in an overstating narrative, which is a potential weakness.
> > >
> > > Regarding the comparison between this work and prior literature, there might also be an overstatement in the rebuttal:
> > >
> > > > The work [4] is fundamentally different from ours, both technically and conceptually.
> > >
> > > I can agree that [4] is different from this paper conceptually. But I find the strong claim that "[4] is fundamentally different technically" unconvincing. The data distribution appears similar, which led me to guess the technical setup is similar. Upon further reviewing some derivations in the appendix, I note that "Early Alignment" and "Decoupled Autonomous Systems" sections in the appendix are indeed similar to "Phase I: Dynamics Near Zero" and "Phase II: Near-Two-Neuron Dynamics" in [4]. These similarities suggest a closer technical relationship than the authors' claim of "fundamentally different".
> > >
> > > While I appreciate the authors' efforts in addressing the review, at this stage I am inclined to keep my original score.

---

> > > > ### Author Response · Authors · 2024-11-26
> > > >
> > > > We thank the reviewer for their answer and expressing their concerns.
> > > >
> > > > > Overstating narrative
> > > >
> > > > The main narrative of our work is that “in recent empirical studies, interpolation is not achieved at the convergence point of training after the *optimization threshold*. Instead, the parameters reach an estimator that is not a global minimum of the training loss, but does generalize well to new data”.
> > > > This is illustrated by recent works on ICL and diffusion, where this observation is made clear but we never claim it only holds in these settings.
> > > >
> > > > We have reformulated the introduction to better address the limitations raised by the reviewer (see Line 078 of the revision). Note that, in the original submission, we had already clarified in the abstract (and title) that our work focuses exclusively on shallow ReLU network architectures. Furthermore, in the Discussion section, we explicitly stated that we consider a simplistic network architecture and emphasized that investigating more complex architectures, such as those with GeLU activations or attention layers, would be valuable future work. We hope this addresses the reviewer’s concerns.
> > > >
> > > >
> > > > > Overstatement in the rebuttal
> > > >
> > > > Indeed, the analysis of [4] and our Theorem 2 both rely on a first, early alignment phase (which is known to be present in both regression and classification, see e.g., [A,B,C,D]) and a second, “near two neurons” phase. We added in the revised version (line 385) that our analysis of the neurons growth also follows a structure similar to [4] (we only wrote it was similar to [C] in the first version).
> > > >
> > > > Yet, we do not think this decomposition of the dynamics justifies a strong technical similarity between both works, as it is also shared by numerous works in the literature (notably, [ B, C, E, F, G, H, I]).
> > > >
> > > > More precisely, the main technical differences between our proof of Theorem 2 and Lyu et al (2021) are the following ones:
> > > > - in our work, the initialization scale is independent of both the width $m$ and the sample complexity $n$ (after some threshold), while their main result holds for both fixed $m$ and $n$. Thus, our theorem can even be extended to the infinite width, mean field limit case, besides holding while $n\to\infty$. This requires a finer control of the dynamics **at each phase**
> > > > - a main difference is in the last phase. While they use an asymptotic rate argument, typical of classification, we instead rely on a local PL argument, typical of regression
> > > > - their analysis requires a non-branching point assumption, that does not generally hold (see their Section I.2.1)
> > > >
> > > > Additionally, our “decoupled autonomous system” argument is unrelated to the “exact/approximate embedding” argument of Lyu et al (2021). Here we show that after the first phase, we can consider a different ODE (which considers linear activations on a fixed subset of the data) and that its solution will **exactly** coincide with the solution of the original ODE (differential inclusion) with ReLU activations. We did not encounter such a technique in the literature.
> > > >
> > > > In opposition, the “exact/approximate embedding” couples the parameters with another solution of the same ODE, which is close to the true parameters after the early alignment phase. Both arguments are needed for very different reasons: ours to control separately positive and negative neurons; theirs to show the two-neuron-dynamics (which corresponds to our Phases 2 and 3).
> > > >
> > > > More generally, our work relies on a phase decomposition of the training dynamics that is quite usual in the literature (again [4, B, C, E, F, G, H, I]).
> > > > While we do not claim this decomposition to be a novelty in our work, there is currently no known general argument to control the complete dynamics (especially if we want the results to hold for arbitrary values of $m$ and $n$). We thus require to analyse the training dynamics nearly from scratch.
> > > >
> > > > We want to insist again that our Theorem 2 is the first to our knowledge that provides such a detailed analysis of the training dynamics, while being simultaneously:
> > > > - valid for arbitrarily large values of $n$
> > > > - converging towards a spurious local minimum of the training loss
> > > >
> > > > For all these different reasons, we believe our work is technically different from [4] and hope you would agree with that. If the reviewer feels needed, we can also add this comparison in the revised version of the paper.
> > > >
> > > > Lastly, if the reviewer has any writing suggestion to reduce this “overstatement effect”, we would be glad to add them in the revised version of the paper.

---

> > > > > ### Author Response · Authors · 2024-11-26
> > > > >
> > > > > ## References
> > > > >
> > > > > [A] Maennel, H., Bousquet, O., & Gelly, S. (2018). Gradient descent quantizes relu network features.
> > > > >
> > > > > [B] Boursier, E., & Flammarion, N. (2024). Early alignment in two-layer networks training is a two-edged sword.
> > > > >
> > > > > [C] Tsoy, N., & Konstantinov, N. (2024). Simplicity Bias of Two-Layer Networks beyond Linearly Separable Data.
> > > > >
> > > > > [D] Kumar, A., & Haupt, J. (2024). Directional convergence near small initializations and saddles in two-homogeneous neural networks.
> > > > >
> > > > > [E] Chistikov, D., Englert, M., & Lazic, R. (2023). Learning a neuron by a shallow relu network: Dynamics and implicit bias for correlated inputs.
> > > > >
> > > > > [F] Min, H., Mallada, E., & Vidal, R. (2023). Early neuron alignment in two-layer relu networks with small initialization.
> > > > >
> > > > > [G] Wang, M., & Ma, C. (2024). Understanding multi-phase optimization dynamics and rich nonlinear behaviors of relu networks.
> > > > >
> > > > > [H] Boursier, E., Pillaud-Vivien, L., & Flammarion, N. (2022). Gradient flow dynamics of shallow relu networks for square loss and orthogonal inputs.
> > > > >
> > > > > [I] Bui Thi Mai, P., & Lampert, C. (2021). The inductive bias of ReLU networks on orthogonally separable data.

---

> > > > > > ### Comment · Reviewer_uxcs · 2024-11-27
> > > > > >
> > > > > > The proposed edits help address my overstatement concern. Thank you for taking the time to respond and revise the paper. I have read your replies and also given the paper a more careful read.
> > > > > >
> > > > > > I tried to work through some math in the paper to get a better intuition of what's going on. In doing so, I seem to have encountered a major issue: I suspect that the trained models actually go to the global minima of training loss for a large number of training samples, rather than spurious local minima as the paper says. Here's why.
> > > > > >
> > > > > > As proved by [1], two-layer ReLU networks without bias terms (the same model used in this paper) cannot represent any nonlinear function with odd frequencies. Thus what the model can represent is $h(x)=w^\top x + g(x)$ in which $g(x)$ is a function with only even frequencies, i.e. satisfying $g(x)=g(-x)$. If we plug this and the dataset defined in Eq(7) of the paper into the training loss, we get
> > > > > > $$
> > > > > > \begin{align}
> > > > > > L &= \sum_{k\in[n]} (w^\top x_k + g(x_k) - \beta^\top x_k - \eta_k)^2
> > > > > > \end{align}
> > > > > > $$
> > > > > > For a large $n$, many terms go to zero and what we are left with is
> > > > > > $$
> > > > > > L = \sum_{k\in[n]} (w - \beta)^\top x_k x_k^\top (w - \beta) + g(x_k)^2 + \eta_k^2
> > > > > > $$
> > > > > > The global minimum of this training loss is attained when $w=(XX^\top)^{-1}XY, g(x_k)=0$. This contradicts the core mechanism of optimization threshold described in this paper: "trained models goes from global minima to spurious local minima of the training loss as the number of training samples becomes larger than some level" (quoting the abstract).
> > > > > >
> > > > > > So my current understanding of the phenomona presented in this paper is that for any $n$, small or large, the model converges to the global minima of training loss. The reason why the model trained with a larger dataset doesn't intepolate the noise and generalizes better is that: the model used in this paper simply cannot represent the intepolating solution anymore for large $n$. Consequently, the observation of optimization threshold in this case of two-layer ReLU networks without bias is more an artifact of the unfortunate choice of having used a model with inherent expressivity limitation, which seems to be a much less interesting explanation than the "global min -> suprious local min" transition proposed in the paper.
> > > > > >
> > > > > > If my analysis is wrong, I welcome correction. If my understanding is not wrong, I'm sorry that I'll have to vote for rejection. Nonetheless, I thank the authors for their manuscript and dedicated responses, which have let me think about this problem more deeply.
> > > > > >
> > > > > > [1] Ronen et al. (2019). The convergence rate of neural networks for learned functions of different frequencies. NeurIPS.

---

> ### Author Response · Authors · 2024-11-28
>
> We thank the reviewer for their interest with our work and the opportunity to clarify these points.
>
> First, we note that as long as the $(x_i)$ are pairwise non-proportional, any dataset $(x_i,y_i)$ can be fitted by a one-hidden-layer network with ReLU activation, even without a bias term. This property follows directly from the positivity of the NTK matrix (see, e.g.,  [A] Theorem 2 for the general case, and [B] Corollary 3.1 for data restricted to the sphere). Importantly, for ReLU activation, this memorization property does not require bias terms, in contrast to universal approximation.
>
> Second, we agree with the reviewer that, in the limit of infinite samples, the global minimizer of the training loss (which coincides with the test loss) corresponds to the Bayesian estimator, which is linear in our data model. However, the key focus of our work is on the finite-sample regime, which introduces significant challenges and subtleties.
>
> In this regime, the reviewer’s reasoning does not fully apply because the asymptotic cross terms cannot be neglected when both the sample size $n$ and the network width $m$ are large.
>
> To summarize, the reviewer states that ERM corresponds to linear regression when $n$ is large. While this is true in certain cases, it does not hold when the network width $m$ remains significantly larger than $n$, as interpolation is still possible in this regime. Importantly, the convergence to linear regression in our analysis comes from an optimization problem rather than a representation limitation. This distinction is discussed in the paragraph “Absence of Interpolation.”
>
> We hope this clarification resolves any misunderstandings.
>
> Best regards,
> the authors
>
> ## References
>
>  [A] Carvalho, L., Costa, J. L., Mourão, J., & Oliveira, G. (2024). The Positivity of the Neural Tangent Kernel.
>
> [B] Montanari, A., & Zhong, Y. (2022). The interpolation phase transition in neural networks: Memorization and generalization under lazy training.

---

> > ### Comment · Reviewer_uxcs · 2024-11-28
> >
> > Thank you for providing this explanation. It might make the result and claim more convincing to
> >
> > - Mention that the no bias simplification means the network doesn't have full expressivity (even with infinite width) and discuss whether the phenomona changes when using a standard biased ReLU network.
> > - Mention that the convergence to global minimum or spurious local minimum may not be a dichotomy separated by optimization threshold. As least from our discussion, the transitions include "global min -> spurious local min -> global min" for small, large, infinite $n$.
> >
> > I now have a clearer understanding of the result and have adjusted my review. I have no further comments.

---

> > > ### Author Response · Authors · 2024-11-29
> > >
> > > We thank the reviewer for their valuable feedback.
> > >
> > > We will add the suggested improvements in the revised version (once modifications are authorized again).

---

### Official Review · Reviewer_afoB · 2024-11-02

**Soundness:** 3
**Presentation:** 2
**Contribution:** 2
**Rating:** 5
**Confidence:** 3

**Summary:**

The paper proves a simplicity bias of gradient flow on two-layer ReLU networks
which prevents them from converging to the training loss's global minimum when
the training set is large enough. Instead, the network's neurons align with a
small number of critical directions in the early phase of training, which guides
gradient flow towards the minimum of the population loss, therefore improving
generalization.

**Strengths:**

- **S1 Motivation:** The paper derives a theoretical explanation for the
  empirically well-documented phenomenon that overparameterized nets generalize
  well despite their capability to fit the training data. The simulation data
  seems to be in agreement with the presented theory. This provides solid
  insights into a general phenomenon.

**Weaknesses:**

- **W1 Gap to practical setting:** To achieve tractability, the paper introduces
  assumptions, e.g. that gradient flow governs the training dynamics, and that
  the net uses ReLU activations. In the introduction, it argues that the
  discussed phenomenon is found in a wide class of models (diffusion, language
  models), though. These models usually contain other activation functions, are
  trained with finite step-size SGD, or even other optimizers like Adam(W). The
  paper acknowledges some of these limitations, but does not probe these
  directions empirically. I think it would be very insightful to report the
  behaviour with a different activation function (say GeLU, popular in LLMs), or
  when using a different training algorithm (say AdamW), or when initializing
  the weights with the default scheme.

- **W2 (minor) notation:** I think the notation could at some points be improved
  to make the paper easier to follow, for instance by introducing bold symbols
  for vectors, explicitly specifying the types of some objects (e.g. the
  elements of the set in L153) or operations (e.g. if L143 is an element-wise
  vector multiplication). I also sometimes found the notation a bit confusing,
  e.g. in L177, where $\theta$ contains $w$, so $\theta=0$ should imply $w=0$,
  but $w$ is a separate argument!? Some symbols are undefined, e.g. the
  distribution $B(0,1)$ in L314, and the norm on the LHS in L299. To some
  extent, this limited my understanding of the results, and how they go beyond
  existing works.

**Questions:**

- Q1: Could you elaborate on the optimization threshold's dependency on the
  network's width $m$, and provide an additional experiment similar to Figure 1,
  but using a second value for $m$ (similar to the larger value of $d$ in the
  appendix)? Also, what is $m$ in the network used to generate Figure 1?

- Q2: Definition 1 is not self-contained as it misses the object that possesses
  an extremal vector. This object should be $G_n$. Can you make the connection
  between $G_n$ and $D$ more explicit? Is it possible to visualize the concept
  of an extremal vector in a low-dimensional example?

- L325: In the parenthesis, it should be $S_-$.

---

> ### Author Response · Authors · 2024-11-22
> **Rebuttal to Reviewer afoB**
>
> We thank the reviewer for his insightful feedback. We now answer your different concerns.
>
> > W1 Gap to practical setting
>
> We agree that it is not immediately clear whether the considered setting reflects the behavior of more complex architectures encountered in practice. This challenge is common in the theoretical Deep Learning literature, where analyses are typically conducted on simplified models, though they might still provide insights into more complex settings. Notably, the study of Gradient Flow is widely used in the literature as it serves as a first-order approximation of SGD under small learning rates.
> As mentioned in the experimental details (Appendix A.1), our experiments are run with SGD and a general default initialization scheme. In the revised version, we have added additional experiments with GeLU activations in Appendix A.3 (and will add Adam+GeLU once we run these experiments). We prefer Adam rather than AdamW to follow the experimental setup of Raventos et al. (2024) and because our focus is on implicit regularization, thus avoiding explicit regularization techniques.
>
> The experiments for GeLU (and preliminary observations for Adam) yield similar observations as in our original setup.
>
> > W2 notations
>
> We chose to avoid using bold characters for vectors, as we found that incorporating them made the notation cumbersome, given the constant presence of vectors in our equations.
>
> $\mathfrak{D}_n$ is defined as a function of two separate arguments $(w,\theta)$.
> This definition is intentional, as $\mathfrak{D}_n$ is continuous only in its second argument $\theta$, but not in $w$, even though $w_i(t)$ is included in $\theta(t)$. Note that in L177 (L187 in revised version), $w$ represents an arbitrary vector whereas  $w_i(t)$ and $\theta(t)$ specifically refer to the network parameters.Thus, Definition 1 is mathematically rigorous and states static properties of $\mathfrak{D}_n$, as a function of two arguments.
>
> > Q1: Could you elaborate on the optimization threshold's dependency on the network's width m
>
> As mentioned in the experimental details, the experiments are run with $m=10\ 000$ (this is now clarified in the main text). The optimization threshold in Theorem 2  is independent of the width $m$ (see L 403 in revised version). The strength of our result lies in the fact that it holds for arbitrarily large m, where interpolation remains possible even for very large n.
>
> The only dependency of Theorem 2 with $m$ is through the probability of the result holding, which is of order $1-O(2^{-m})$. It is because we need at initialization that at least one neuron satisfies $a_j>0$ and at least one satisfies $a_j<0$.
>
> > Q2: definition 1 is not self contained
>
> Thanks for this suggestion, we modified the definition to “a vector is said extremal **with respect to G_n**”.
> $\mathfrak{D}_n(\cdot,\mathbf{0})$ corresponds to the sub-gradient of $G_n$ in the new version. Extremal vectors correspond to the KKT points of the maximization/minimization problems of $G_n$ on the unit sphere in $\mathbb{R}^d$.

---

> > ### Comment · Reviewer_afoB · 2024-11-25
> >
> > Thank you for your responses! I appreciate the additional experiments, checked out the new GeLU results in Fig. 3 (Appendix A.3), and will take them into account during the discussion with the other reviewers. For now, I don't have follow-up questions.

---

> ### Author Response · Authors · 2024-12-03
>
> Dear Reviewer afoB,
>
> As the discussion period is ending soon, we wanted to check if you have any additional questions about our paper.
>
> Best regards,
>
> The authors

---

### Official Review · Reviewer_V7Fz · 2024-11-05

**Soundness:** 4
**Presentation:** 2
**Contribution:** 3
**Rating:** 6
**Confidence:** 2

**Summary:**

The paper proves results for a 2-layer ReLU network trained with gradient descent, and its convergence with respect to sample complexity. First, it proves that neuron weights cluster around a few extremal vectors as data size increases. Then, it shows that for a linear problem, the same network converges to the ordinary least squares estimator as data size goes beyond a threshold. This shows that contrary to common wisdom that neural networks interpolate between training data points, neural networks in this setting converge to simpler solutions with sufficient data.

**Strengths:**

Originality: the paper has a similar setting and results as some previous works (Boursier & Flammarion, Tsoy & Konstantinov) but generalizes to more datasets (e.g. beyond XOR) with unbounded sample complexity (e.g. theorem 2).

Quality: I cannot speak too much to the proofs, but they appear to follow from the established methods of Boursier & Flammarion. Despite being theoretically motivated, the paper has some nice experimental supporting evidence, e.g. appendix A.2 which shows the large gradient steps are not sufficient to explain simplicity bias.

Clarity: the discussions and experiments are straightforward to follow.

Significance: the immediate impact of this work is to build on analyses of 2-layer networks and regression settings to strengthen the case for the simplicity bias of neural network optimization. There is potential to explain the good generalization of in-context learning and diffusion models, but much work remains to extend the results to these applications.

**Weaknesses:**

My main concern is that the sample complexity bounds appear loose relative to empirical evidence, even in the regression setting presented in this work. The paper's structure could also be improved.

As discussed by the authors in "Limitations", there is a large (more than cubed) dependence of sample complexity on the input dimensionality, which makes it hard to relate this work to practical tasks, where dimensionality tends to grow more than sample complexity. It would be nice to see the experiment in figure 1 repeated for at least 2 more values of $d$ other than 5 to see if $n^\star$ evolves with respect to dimensionality at a different rate in practice.

Overall the sections are somewhat disjointed due to the way discussions are interleaved with results: it would be easier to follow if every result was presented in a consistent manner (e.g. in order of the idea/intuition, context/prior work, the actual mathematical statement, the implications of the statement). In particular, section 2 can be hard to follow, especially regarding extremal vectors, which are introduced with little explanation or supporting intuition - some of the discussion and context for these concepts occurs after proposition 1. Although space is limited, explaining more details of the notation and key proof concepts would help make the work stand on its own instead of requiring unfamiliar readers to refer to prior work (e.g. intuition for the set $\mathfrak{D}$, reasoning for the domination property of initialization).

Since theorem 2 states neurons should cluster into one of two states, figure 1 could use include some information about the distance between neuron weights and the true OLS data model. For example, the mean cosine similarity after training could be plotted.

**Questions:**

Prior work (Boursier & Flammarion) discusses early phases of training where directional alignment occurs. Could the authors empirically or theoretically discuss if sample complexity influences when this early training phase occurs?

**Details Of Ethics Concerns:**

EDIT: the ethical concern has been resolved with additional citations and clarification in the text.

Section 2.2 and 2.3 follows the organization, variable naming, and in many parts matches word-for-word with section 2 of Boursier and Flammarion (2024). Section 2.1 is also similar to 2.1 in  Boursier & Flammarion. While mathematical definitions cannot be entirely original, the degree of similarity warrants a need to explicitly cite Boursier & Flammarion so that readers can trace the origin of the definitions. Some of the paraphrasing of Boursier & Flammarion may also contribute to making the definitions harder to understand (see Weaknesses).

---

> ### Author Response · Authors · 2024-11-22
> **Rebuttal to Reviewer V7Fz**
>
> We thank the reviewer for his insightful feedback. We now answer your different concerns.
>
> >  the sample complexity bounds appear loose relative to empirical evidence
>
> The theoretical analysis of this (seemingly simple) setting reveals quite intricate, and clearly leads to some looseness in the provided bounds. Yet, our empirical observations confirmed a similar dependency of the sample complexity bound wrt the dimension d. We added preliminary experiments with dimensions 10 in the revised version confirming that trend. A more polished version of these experiments, as well as results for d=20 will be included in future version. Note that these experiments require significant computational resources (see our response to reviewer EBuW).
>
> > the sections are somewhat disjointed due to the way discussions are interleaved with results
>
> Thank you for the suggestion. In the revised version, we added further intuition and explanations about the early alignment phase in Section 2 and reordered Sections 4.1 and 4.2 to improve clarity.
>
> We note that providing more detailed insights into the early alignment phase requires substantial space and has already been addressed in prior works (e.g., Maennel et al., 2018 and Boursier and Flammarion, 2024). Instead, we decided to focus on our own original contributions, including only the tools and understanding of the early alignment phase necessary for our analysis.
>
> > figure 1 could include some information about the distance between neuron weights
>
> In the appendix of the revised version, we have included histograms of the mean cosine similarity with the OLS estimator at the end of training for different values of n. These results confirm the predictions of Theorem 2.
>
> > Could the authors empirically or theoretically discuss if sample complexity influences when this early training phase occurs?
>
> This training phase occurs in a general setting, provided the initialization scale is sufficiently small. After concentration, the sample complexity $n$ has little impact, as the early-phase dynamics are primarily driven by the expected  value $G(w)$. Indeed, the dynamics of this phase depend only on the function $G_n$​, meaning that the sample complexity $n$ influences the dynamics only through its effect on the geometry of $G_n$​. The goal of Section 3 is to study this influence of $n$ on the shape of the function $G_n$​ (or equivalently, the vectors $D_n$​). In the general setting of Section 3, the initialization scale threshold is independent of $n$ for large enough $n$, as $D_n​(w)$ concentrates towards $D(w)$.
>
> > Ethical concerns about similarity with Boursier and Flammarion (2024) in section 2
>
> Our work builds on the results of Boursier and Flammarion (2024) and the goal of section 2 is precisely to summarize these results as the starting point of our analysis. To avoid any additional confusion, we follow the same notations (hence the similarity in Section 2.1) and definitions. We now explicitly state at the beginning of section 2 that we introduce here the results of Boursier and Flammarion (2024). However, the main explanatory paragraphs in Section 2.3 (early alignment and its implications) are novel.

---

> > ### Comment · Reviewer_V7Fz · 2024-11-26
> >
> > Thank you for the rebuttal, as well as the revised presentation and additional figures. I have removed the ethical concern following the citation of Boursier and Flammarion.
> >
> > Regarding some points raised by other reviewers, I do not think that the limited theoretical setting is necessarily an issue since there is a well-established body of literature applying gradient flow to 2-layer ReLU networks, and simplified settings are necessary to better understand fundamental training dynamics. However, I do agree with reviewer uxcs that if the work has similar technical construction to prior work (e.g. Lyu et al. 2021) then it is less impactful. The novel perspective on the results is still interesting and valuable, so I will keep my score.

---

> ### Author Response · Authors · 2024-11-29
>
> We thank the reviewer for their useful feedback.
>
> Now we answered the different concerns of Reviewer uxcs, we would be happy to answer any remaining concern on your side if there is any.

---

### Official Review · Reviewer_EBuW · 2024-11-06

**Soundness:** 3
**Presentation:** 3
**Contribution:** 3
**Rating:** 6
**Confidence:** 4

**Summary:**

This is a theoretical paper seeking to explain the phenomenon that in some situations overparameterized models, once the number of noisy training samples is large, fail to interpolate the training data and instead converge to a solution that minimizes the test loss.  This was observed with in-context learning and with diffusion models.  In this paper, the architecture is overparameterized two-layer ReLU networks, and the focus is on training them from a small initialization by gradient flow on noisy data which is labelled by a linear teacher and which satisfies some further simplifying assumptions.  The main result confirms the motivating phenomenon, and its proof provides further insights, pinpointing an early-phase alignment of neurons as the principal cause.  Another contribution are concentration bounds for the extremal vectors that drive the alignment process, and the assumptions here are less restrictive than in the main result.  The paper also reports and discusses numerical experiments in setups related to but extending the setting of the main theoretical result.

**Strengths:**

The paper is well written and the main result is accompanied by a detailed discussion.

The theoretical results are non-trivial, and their proofs are provided in the appendix.

The concentration bounds result may indeed be useful for future work.

The experimental results are interesting, and their discussion is informative.

**Weaknesses:**

The gap between the theoretical setting in this work and the empirically observed phenomenon with in-context learning and diffusion modes is large, and it is not clear whether and how the properties of the training dynamics identified here are related to what actually happens in those practical situations.  (E.g. it might be that case that the stability issues studied by Qiao et al. in NeurIPS 2024 are at play there to a greater extent than the neuron alignment in early training.)

As far as I can see, equation (8) is actually an assumption in Theorem 2, however that is not stated in Theorem 2, and moreover equation (8) is presented in the middle of a paragraph which starts with an example.  It would be clearer to have equation (8) as Assumption 2.

More minor points:
- The notation B(0, 1) is not defined, and presumably the 0 there should be bold because it is a vector.
- It seems to me you wanted to say that the data points are the columns of X, given how you multiply X and Y in Theorem 2.
- Perhaps stick to either transpose or angular parenthesis as the notation for inner products.
- In the Experiments section, it could be informative to state what m was.
- In the References, check whether papers cited as on arXiv have been published, and consider providing a clickable link for each item.

**Questions:**

Can you say more about your conjecture of order n effective width at the end of training in the orthogonal training data setting?  As far as I understand, Boursier et al. in NeurIPS 2022 showed that this width would be only 2, one group of aligned neurons for the positive labels, and another for the negative labels?

How essential is the ReLU activation for the main result?

How much compute was needed for the experiments?

---

> ### Author Response · Authors · 2024-11-22
> **Rebuttal to Reviewer EBuW**
>
> We thank the reviewer for his positive and insightful feedback. Additionally, we thank you for all the suggestions of rewriting that have been incorporated in the revised version. We now answer your different concerns.
>
> > gap between theoretical setting and empirically observed phenomena
>
> The exact reason for the observed phenomena in practice remains unclear. It could stem from the stability issue raised by Qiao et al., the problem of convergence toward local minima, a combination of these factors, or even more complex considerations arising from more complex data structures. The goal of our work is to propose a plausible  theoretical explanation for the observations in ICL and diffusion models. Validating whether any of these factors fully explain the empirical observations would require extensive experiments, which we leave for future work. Notably, we could consider the ICL experiment of Raventos et al. (2024) with significantly smaller learning rates.
>
> > Conjecture of effective width in the orthogonal training data setting
>
> In the work of Boursier et al., the effective width is only 2 at the end of training. However, achieving this minimal width requires a significantly overparameterized network, with roughly $m\gtrsim 2^n$. This overparameterization is necessary to ensure that, at initialization, there exists at least one neuron $w_i$ positively correlated with all data points $x_k$ such that $y_k>0$, and similarly, another neuron $w_j$ positively correlated with all negative-labeled data points.
>
> Studying their orthogonal data setting under mild overparameterization (eg $m=poly(n)$) remains an open problem, as the initialization condition would no longer hold in such cases. Consequently, multiple neurons would be required to fit all positive labels (and similarly for negative ones). Preliminary investigations suggest that this scenario would lead to an effective width that increases with $n$. However, determining the exact dependency on $n$ would require further analysis.
>
> > How essential is the ReLU activation
>
> Homogeneous activations (e.g., ReLU or leaky ReLU) are crucial to the analysis of the early alignment phase and, more generally, to many analyses that allow an arbitrarily large or infinite width. While our work could be directly extended to any leaky ReLU activation, we focused on ReLU for the sake of clarity
>
> When considering more general activations (e.g. differentiable ones), the early alignment phase still happens (see Kumar and Haupt 2024), yet it does not break omnidirectionality of the neurons as $m\to\infty$. In consequence, infinitely large networks with differentiable activations should still interpolate at the end of training (see Chizat and Bach 2018). Nevertheless, achieving this interpolation in practice often requires very large $m$ in practice, as interpolation of the data seems even harder to reach with GeLU activations as suggested in our new experiments.
>
> > How much compute was needed
>
> The experiments were run on a personal computer (MacBook pro) and required approximately 100 hours of compute. This relatively long time is specifically due to the fact that we considered largely overparametrized networks and long training times, to ensure that convergence of the parameters was reached. The precision about the required compute has been added in the revised version of the paper.

---

> > ### Comment · Reviewer_EBuW · 2024-11-22
> >
> > Thank  you for the reponse!

---

### Author Response · Authors · 2024-11-22
**Rebuttal**

We thank the reviewers for their insightful feedback. All of their comments have been taken into account in the revised version of the paper. We here answer concerns that were shared by several reviewers and answer specifically to individual reviews below.

# Relevance of the theoretical setting to the practice.

It is challenging to assess how well the considered setting reflects the behavior of more complex architectures encountered in practice. This limitation is not unique to our work but is a common aspect of the Deep Learning theory literature, where analyses are often constrained to simplified models for tractability. Despite this simplicity, such settings are valuable as they enable the illustration of non-trivial phenomena that might also arise in more complex scenarios—a significant result in its own right.
The study of Gradient Flow is widely adopted in the literature because it serves as a first-order approximation of both Gradient Descent (GD) and Stochastic Gradient Descent (SGD) under small learning rates, facilitating simpler and more interpretable analyses.
Reviewers have also noted that the observed phenomena in practice may stem from various factors: the convergence issues towards local minima described in our work, the stability issues highlighted by Qiao et al., or even more complex factors such as the use of attention layers. In this paper, we propose a plausible explanation and rigorously study it from a theoretical perspective. Determining the precise cause of these practical phenomena requires further investigation, as extensive experiments would be necessary to determine which of these factors explains the empirical observations. In response to the reviewers' suggestions, we have added new experiments in the appendix of the revised version and added a discussion on these potential explanations.

# About writing clarity
We thank the reviewers for their valuable suggestions regarding the writing. We have uploaded an updated version on OpenReview, incorporating all your comments. Additionally, we provided a LaTeXDiff PDF as supplementary material to highlight the changes compared to the previous version.

Specifically, in response to the reviewers’ feedback, we have:
- Switched the discussion and sketch of proof sections,
- Added preliminary experiments in the appendix as suggested,
- Included explanations about the early alignment phase (Section 2), and
- Addressed all minor comments, including typos.

---

### Author Response · Authors · 2024-11-25

Dear Reviewers,

As the discussion period is ending soon, we wanted to check if our reply addressed your different concerns or if you have additional requests.

Additionally, we also just added new experiments in the appendix, considering GeLU activation + Adam optimizer simultaneously.

Best regards,

The Authors

---

### Meta-Review · Area_Chair_mf57 · 2024-12-23

**Metareview:**

This paper studies the phenomenon that networks often converge toward simpler solutions rather than interpolating the training data. In particular, it was observed earlier that gradient descent converges to a simpler solution closely related to the true loss minimizer (rather than converging to the global minimum of the training loss). The paper examines this phenomenon for 2-layer ReLU networks.

The reviews for the paper are very much on the borderline. While the reviewers found the work to be somewhat interesting, there were concerns regarding the practical implications of the work, gap in theoretical work & practical settings, somewhat unclear presentation and unclear relation with prior works. Some of these concerns were addressed by the authors during the rebuttal. However, the concerns the practical implications are very valid and have to be addressed carefully.  Also, the concerns about prior works are valid. While addressed to some degree by the authors, they still failed to convince the reviewers completely. I think it is important to address these concerns carefully before acceptance. I recommend rejection in the current form.

**Additional Comments On Reviewer Discussion:**

The authors did a great job improving the clarity in the paper. While some of the concerns regarding the practicality of the approach and relation with prior works have somewhat been addressed, they failed to completely convince the reviewers. I think these are valid concerns from the reviewers and need to addressed carefully before acceptance.

---

### Decision · Program_Chairs · 2025-01-22

Reject